

# How warm was Greenland during the last interglacial period?

Amaelle Landais[1], Valérie Masson-Delmotte[1], Emilie Capron[2,3], Petra .M. Langebroek[4], Pepijn. Bakker[5], Emma J. Stone[6], Niklaus Merz[7], Christoph C. Raible[7], Hubertus Fischer[7], Anaïs Orsi[1], Frédéric Prié[1], Bo Vinther[2], Dorthe Dahl-Jensen[2].

5    [1] Laboratoire des Sciences du Climat et de l'Environnement - IPSL, UMR 8212, CEA-CNRS-UVSQ-Université Paris Saclay, Gif sur Yvette, France

[2] Center for Ice and Climate, Niels Bohr Institute, University of Copenhagen, Juliane Maries Vej 30, 2100 Copenhagen Ø, Denmark.

[3] British Antarctic Survey, High Cross Madingley Road, Cambridge CB3 0ET, UK

10   [4] Uni Research Climate, Bjerknes Centre for Climate Research, Allégaten 55, 5007 Bergen, Norway

[5] College of Earth, Ocean and Atmospheric Sciences, Oregon State University, USA

[6] BRIDGE, School of Geographical Sciences, University of Bristol, Bristol, UK

[7] Climate and Environmental Physics, Physics Institute, and Oeschger Centre for Climate Change Research, University of Bern, Sidlerstrasse 5, 3012 Bern, Switzerland





**Abstract.** The last interglacial period (LIG, ~129-116 thousand years ago) provides the most recent case study for multi-millennial polar warming above pre-industrial level and a respective response of the Greenland and Antarctic ice sheets to this warming, as well as a test bed for climate and ice sheet models. Past changes in Greenland ice sheet thickness and surface temperature during this period were recently derived from the NEEM

ice core records, North-West Greenland. The NEEM paradox has emerged from an estimated large local warming above pre-industrial level (7.5 ± 1.8°C at the deposition site 126 ka ago without correction for any overall ice sheet altitude changes between the LIG and pre-industrial) based on water isotopes, together with limited local ice thinning, suggesting more resilience of the real Greenland ice sheet than shown in some ice sheet models. Here, we provide an independent assessment of the average LIG Greenland surface warming

using ice core air isotopic composition ($\delta^{15}N$) and relationships between accumulation rate and temperature. The LIG surface temperature at the upstream NEEM deposition site without ice sheet altitude correction is estimated to be warmer by +7 to +11°C (+8°C being the most likely estimate according to constraints on past accumulation rate) compared to the pre-industrial period. This temperature estimate is consistent with the 7.5±1.8°C warming initially determined from NEEM water isotopes. Moreover, we show that under such warm

temperatures, melting of snow probably led to a significant firn shrinking by ~ 15 m. Climate simulations performed with present day ice sheet topography lead to much smaller warming but larger amplitudes (up to 5°C) can be obtained from changes in sea ice extent and ice sheet topography. Still, ice sheet simulations forced by 5°C surface warming lead to large ice sheet decay that are not compatible with existing data. Our new, independent temperature constrain therefore reinforces the NEEM paradox.

**1 Introduction**

Understanding the magnitude, timing and rate of contributions of the Greenland and/or Antarctic ice sheets to the estimated 5 to 10 m increase in global mean sea level during the last interglacial period (hereafter LIG, 129-116 thousand years before 1950, hereafter ka) and therefore ice sheet vulnerability to multi-millennial polar warming remains challenging (Masson-Delmotte et al., 2013; Dutton et al., 2015). Therefore, constraints on past

polar climate and ice sheet response are required. Additionally, polar temperature reconstructions provide a benchmark to assess the ability of climate models in capturing feedbacks which amplify the impact of orbital forcing on polar temperatures (Masson-Delmotte et al., 2011; Otto-Bliesner et al., 2013a; Capron et al., 2014). This latter is also relevant for future climate projections.

Since the 1960s, numerous Greenland deep ice core records have provided evidence for layers of ice located

near bedrock characterised by high values of water stable isotopes ($\delta^{18}O_{ice}$), well above pre-industrial Holocene levels (Johnsen et al., 1997). The climate interpretation of the first records was limited due to poor preservation of deep samples (Camp Century, Dye 3), and the lack of remaining air content preventing any dating by synchronisation with global atmospheric records (i.e. atmospheric $\delta^{18}O$ of $O_2$, hereafter $\delta^{18}O_{atm}$ and $CH_4$) from undisturbed Antarctic records. This synchronisation method was applied for the LIG interval at Summit, where

ice from the LIG was unequivocally identified although not unambiguously datable, but sharp variations in $\delta^{18}O_{ice}$ at GRIP and GISP2 were attributed to stratigraphic disturbances (Grootes et al., 1993; Landais et al., 2004; Landais et al., 2003; Suwa et al., 2006). At NGRIP, continuous climatic and environmental records cover





the last 123 ka (NorthGRIP-community-members, 2004). The Greenland record was recently extended back to 128 ka thanks to a 80 m segment of ice in stratigraphic order found in between disturbed layers at the bottom of the NEEM ice core (NEEM comm. members, 2013). The chronology of this core was tied to an Antarctic ice core age scale, based on common changes in atmospheric composition. The unequivocal matching between the

NEEM LIG layer and the Antarctic $\delta^{18}O_{atm}$ records rules out stratigraphic disturbance (NEEM comm. members, 2013).

Changes in NEEM air content and $\delta^{18}O_{ice}$ were corrected for elevation changes due to the upstream displacement of the deposition site, and combined to infer changes in ice sheet topography, and changes in surface air temperature (NEEM comm. members, 2013). This requires assumptions on NEEM $\delta^{18}O_{ice}$ –

temperature relationships. While Greenland snow isotopic composition has long been related to temperature due to Rayleigh distillation associated with cooling along air mass pathways (Dansgaard et al, 1964), it has been increasingly documented that $\delta^{18}O_{ice}$-temperature relationships are neither stable in time nor in space (e.g. Jouzel et al., 1999) primarily due to changes in the precipitation intermittency, but also evaporation conditions and atmospheric transport (e.g. Krinner et al., 1997; Masson-Delmotte et al., 2011).

The initial LIG temperature estimate (NEEM comm. members, 2013) was performed using the average Holocene $\delta^{18}O_{ice}$-temperature relationship established from other central Greenland ice cores through calibration against borehole temperature at 0.5‰.°C⁻¹ (Vinther et al., 2009). This relationship was also explored in simulations using isotopically enabled atmospheric general circulation models for climate conditions warmer than pre-industrial, either in response to increasing $CO_2$ concentration in projections, or in response to changes

in orbital forcing. These models produced slopes varying from 0.3 to 0.7‰.°C⁻¹ in Greenland, depending on changes in moisture sources driven by changes in sea ice and sea surface temperature patterns (Masson-Delmotte et al., 2011; Sime et al., 2013). Based on these lines of evidence, a slope varying from 0.4 to 0.6 ‰ per °C was used to estimate the range of changes in LIG temperature based on NEEM $\delta^{18}O_{ice}$. At 126 ka, and at the location of the initial snowfall deposition site (about 205 ± 20 km upstream of the current NEEM site),

$\delta^{18}O_{ice}$ was estimated at 3.6‰ above local pre-industrial level, which translated into local surface air temperature warming of 7.5±1.8°C. After accounting for upstream effects and for Greenland ice sheet elevation change based on air content, this led to an estimate of a 8±4°C warming at the NEEM site at 126 ka and about 6±3°C at 122 ka (NEEM comm. members, 2013). In parallel, ice sheet simulations forced by different LIG climate scenarios were investigated to select only those compatible with limited change in ice thickness at

NEEM, based on air content data. This implied limited Greenland ice sheet deglaciation, with a contribution of 1.4 to 4.3 m to the LIG sea level increase (Masson-Delmotte et al., 2013).

These results led to the "NEEM paradox", where the Greenland ice sheet appears resilient to large multi-millennial surface warming. This paradox was further enhanced by the difficulty of coupled ocean-atmosphere climate models to capture such warming (Otto-Bliesner et al., 2013b; Capron et al., 2014), even during the

warmest summer months (van de Berg et al., 2013), and by the inconsistency of Greenland retreat simulated by ice sheet models in response to such warming (e.g. Stone et al., 2013; Helsen et al., 2013). When accounting for a reduced Greenland ice sheet and a retreat in sea ice cover in the Nordic Seas, atmospheric simulations can explain up to 5°C annual mean warming with respect to pre-industrial (Merz et al., 2014a; 2016). Moreover, all LIG climate modelling studies cited above strongly enhance summer precipitation seasonality in Greenland,



suggesting a summer bias for LIG $\delta^{18}O_{ice}$ and weaker annual mean change than the initial estimate of 8±4°C (Masson-Delmotte et al, 2011; Merz et al., 2014b).

Recently, new information on climatic controls on NEEM $\delta^{18}O_{ice}$ has emerged from present-day water isotope monitoring and multi-decadal trends from shallow ice cores (Steen-Larsen et al., 2011; Steen-Larsen et al.,

2014). All these datasets coherently document a surprisingly large present-day response of NEEM $\delta^{18}O_{ice}$ to temperature, with a slope of [0.8-1.2] ‰.°C$^{-1}$ (Masson-Delmotte et al., 2015). If relationships established from the intra-seasonal to the multi-decadal scale remain valid for earlier warm periods such as the LIG, it also implies that the initially reconstructed temperature change based on NEEM $\delta^{18}O$ was overestimated.

Here, we present new, independent information on LIG annual mean temperature change for several Greenland

drilling sites, using the ice core air isotopic composition $\delta^{15}N$. These Greenland records are described in Section 2. Section 3 details the temperature reconstructions with their associated uncertainties, with a focus on the NEEM deposition site. These temperature estimates depend on assumptions on the past relationship between temperature and accumulation rate. Section 4 presents a comparison to modeling outputs for discussion before the conclusions.

**2 Water and air isotope records of the last interglacial in Greenland**

**2.1 Records of water stable isotopes from multiple ice cores on a coherent chronology**

Figure 1 shows the compilation of the LIG $\delta^{18}O_{ice}$ records from NGRIP, GRIP and GISP2 sites on a coherent timescale. NEEM $\delta^{18}O_{ice}$ is presented on a parallel depth scale adjusted for the alignment of $\delta^{18}O_{atm}$ records over the LIG section. As $CH_4$ and $\delta^{18}O_{atm}$ are globally well-mixed atmospheric tracers, comparable values are

measured in the Greenland and Antarctic ice cores at the same time period, accounting for the $CH_4$ interpolar gradient leading to slightly higher $CH_4$ levels in Greenland than in Antarctica (e.g. Dällenbach et al., 2000). The synchronization between the records is therefore based on parallel large variations in $CH_4$ and $\delta^{18}O_{atm}$ from measurements in the air trapped in bubbles. For the end of the LIG and the glacial inception, NGRIP records were placed on the AICC2012 timescale (Bazin et al., 2013; Veres et al., 2013) using $CH_4$ and $\delta^{18}O_{atm}$ tie-points

between NGRIP and the Antarctic EPICA Dronning Maud Land (EDML) ice core (Capron et al., 2010). However, the AICC2012 NGRIP chronology is limited since (1) no synchronization points are available for ages older than 118 ka (supplementary online material in Bazin et al., 2013; Veres et al., 2013) and (2) the mean $CH_4$ level is significantly higher at NGRIP than in the EPICA Dome C (EDC) record (Capron et al. 2012). The latter is interpreted to reflect a strong increase in the inter-hemispheric $CH_4$ gradient, which complicates the

alignment of NGRIP and EDC $CH_4$ records (Capron et al., 2012). Additionally, a slight mismatch is observed between the LIG NGRIP and the recently published EDC $\delta^{18}O_{atm}$ records (Figure 1; Landais et al., 2013), suggesting that NGRIP $\delta^{18}O_{ice}$ may be too young by 2 ka at 121 ka when compared to the other ice core records on the AICC2012 chronology.

The dated GRIP and GISP2 $\delta^{18}O_{ice}$ records are discontinuous, because of strong stratigraphic disturbances over

the bottom 300 m of these Summit ice cores. They were initially placed on the Vostok GT4 timescale (Petit et



al., 1999) using identification of $\delta^{18}O_{atm}/CH_4$ pairs and taking into account the inter-polar $CH_4$ gradient (Landais et al., 2003; Suwa et al., 2006). Here, we have transferred these $\delta^{18}O_{ice}$ records on AICC2012 using the correspondence between the Vostok GT4 and AICC2012 chronologies (Figure 1).

Finally, the LIG section of NEEM can only be dated using $\delta^{18}O_{atm}$ because its $CH_4$ record is contaminated by in-situ production, in relationship with local summer melt during the LIG (NEEM comm. members, 2013). Figure 1 displays the NEEM $\delta^{18}O_{ice}$ record on its depth scale between 2350 and 2490 m, where the linear alignment of depth with AICC2012 is based on the resemblance between EDC and NEEM $\delta^{18}O_{atm}$ records.

The continuous NEEM section spanning the LIG ends just after 128 ka (on the AICC2012 timescale). Indeed, the characteristic abrupt increase of $CH_4$ and high $\delta^{18}O_{atm}$ level identified in Antarctic records at 128 ka is absent from the record. This reveals that the NEEM ice core does not encompass any ice from the penultimate deglaciation at that point, similar to GISP2, GRIP or NGRIP (Figure 1). Whether this hiatus arises from the disappearance of this layer due to melt under warm early LIG conditions or due to specific thinning and flow associated with different physical properties of glacial versus transition ice remains to be fully assessed.

## 2.2 NEEM air $\delta^{15}N$ record

Relative to the free atmosphere mean value, the $\delta^{15}N$ value in air trapped in ice cores is influenced by gravitational fractionation directly related to temperature and to the depth at which bubble lock-in occurs, where the latter is controlled by local mean annual temperature and accumulation rate. In Greenland, changes in firn lock-in depth (LID) can quantitatively be related to changes in surface accumulation and temperature using firn densification models (Goujon et al., 2003; Guillevic et al., 2013; Kindler et al., 2014): an increase in temperature leads to a decrease of the LID because of faster metamorphism, while higher accumulation rates lead to an increase of the LID. During rapid surface temperature changes (e.g. Dansgaard-Oeschger events), $\delta^{15}N$ is also influenced by thermal fractionation (Severinghaus et al., 1998). However, no rapid $\delta^{18}O_{ice}$ changes are found during the LIG, and overall stable NEEM $\delta^{15}N$ values are also coherent with gravitational fractionation occurring under stable surface accumulation rate and temperature (Figure 2).

The single exception is a negative spike recorded at 2384 m depth, which coincides with the strongest $CH_4$ spike, as well as a negative excursion of the $^{10}Be$ record (Sturevik-Storm et al., 2014). We suggest that this singular event reflects positive surface temperatures, leading to intense surface melt and large in-situ $CH_4$ production (Orsi et al., 2015). Firn air transport and thus $\delta^{15}N$ are not expected to be significantly affected by melt layers at the surface (Keegan et al., 2014). This probably explains why most of the $CH_4$ spikes are not associated with any changes in the $\delta^{15}N$ signal. Still, for the negative 0.07‰ $\delta^{15}N$ excursion at 2384 m (corresponding to ~121 ka on the NEEM LIG age scale), we propose that positive surface temperatures have led to a sudden shrinking of the firn by about 15 m using the expression of $\delta^{15}N$ gravitational settling in the firn column:

$$\delta^{15}N = \exp\left(\frac{g \times LID}{R \times T_{mean}}\right) - 1 \cong \frac{g \times LID}{R \times T_{mean}} \qquad (1)$$





where *g* is acceleration due to gravity, *R* the ideal gas constant and $T_{mean}$ (K) the mean temperature of the firn when bubbles are isolated.

For the rest of the LIG our $\delta^{15}$N record shows only very subtle changes and we take the mean $\delta^{15}$N value of 0.251 ‰ over the time interval 122-126 ka to obtain a representative mean $\delta^{15}$N value (Table 1).

## 2.3 Spatial structure of isotopic anomalies

To assess the spatial extent of the isotopic anomalies, the magnitude of $\delta^{15}$N are compared to the water stable isotope ($\delta^{18}O_{ice}$ and deuterium excess, hereafter d-excess) anomalies recorded in different Greenland ice cores between the LIG and pre-industrial (Table 1).

For $\delta^{18}O_{ice}$, the strongest increase from pre-industrial to the LIG is recorded at Summit (+3.4 ‰) and NGRIP (+3.3 ‰), and the smallest increase at NEEM (+ 2.1 ‰). However, the NEEM anomaly must be corrected for upstream effects: due to ice flow, the LIG ice at NEEM originates from a 330 m higher upstream location (at the south-east of NEEM) where mean annual $\delta^{18}$O today is approximately -35 ‰ (NEEM comm members, 2013). At this deposition site, the LIG isotopic anomaly is therefore 3.6 ‰, close to the value at Summit. We conclude that changes in $\delta^{18}O_{ice}$ are homogeneously ~ 3.2-3.6 ‰ higher at the LIG than at pre-industrial in central and north-west Greenland (Table 1).

d-excess is not significantly different between the LIG and pre-industrial period at the different drilling sites listed on Table 1. Pre-industrial d-excess values are also very similar among these different sites. This is the reason why extrapolating the surface d-excess values in the NEEM-NGRIP regions enables us to estimate the pre-industrial d-excess at the upstream NEEM deposition site to a conservative value of 11‰.

Assuming that no abrupt climate change took place from 122 to 126 ka (a hypothesis supported by the relatively flat NEEM $\delta^{18}O_{ice}$), changes in mean $\delta^{15}$N are expected to reflect changes in LID. The spatial structure of the $\delta^{15}$N changes differs from the pattern of $\delta^{18}O_{ice}$. Indeed, the smallest $\delta^{15}$N increase is observed at NGRIP (-0.02 ‰) (but the record ends at 123 ka) and the largest one at GRIP and GISP2 (-0.07 to -0.09 ‰), with an intermediate signal at NEEM (-0.04 ‰), albeit with an uncertainty here due to the lack of data for the NEEM deposition site today. Because of stratigraphic disruptions, no continuous record is available at Summit. We thus cannot exclude that low $\delta^{15}$N levels observed at GRIP and GISP2 on (122-126 ka) ice sections reflect a temporary $\delta^{15}$N decrease caused by thermal fractionation or firn shrinking as for the NEEM $\delta^{15}$N value at 2384 m (121 ka). We therefore note regional differences for the different available datasets, but stress their heterogeneities (time span, discontinuity, and lack of present-day reference) preventing any robust conclusion.

## 3 Temperature reconstructions

### 3.1 Reconstructions based on $\delta^{18}O_{ice}$

Today, NW Greenland accumulation is biased towards summer precipitation (based on regional and general circulation atmospheric models; Steen-Larsen et al., 2011). NEEM summer $\delta^{18}O_{ice}$ was monitored through continuous measurements of surface water vapor isotopic composition in 2010-2012 (Steen-Larsen et al., 2013;Steen-Larsen et al., 2014), revealing an intra-annual $\delta^{18}O_{ice}$ –temperature slope of 0.85 ‰ per °C at the



intra-summer scale. From 1979 to 2007, the increasing trend of $\delta^{18}O_{ice}$ recorded in NEEM shallow ice cores was scaled to simulated and estimated local surface air temperature trends, resulting in a multi-decadal slope of 1.05±0.2 ‰.°C$^{-1}$ for warming above pre-industrial conditions (Masson Delmotte, 2015). These various estimates suggested that the average Holocene $\delta^{18}$O-temperature relationship of 0.5‰.°C$^{-1}$ based on the calibration with

borehole temperature data at other Greenland ice core sites (Vinther et al., 2009; Vinther et al., 2010) may not be valid for NEEM. Differences between Greenland locations are expected due to changes in the seasonality of precipitation (summer bias at NEEM but not in central or south Greenland), moisture origin as well as possible changes in boundary layer stability and relationships between surface and temperature relationship.

Using the present-day temporal slope detailed above, the LIG $\delta^{18}O_{ice}$ anomaly at NEEM deposition site

translates into a warming of 2.9-4.2°C, two times smaller than the initial estimate based on Holocene calibrations for other sites. Still, it is difficult to assess whether the present-day calibration can apply for the LIG, marked by a different orbital forcing than today, likely with a reduced sea ice extent and different moisture transport pathways (Sime et al., 2013). The second-order isotopic parameter, the d-excess, can provide information on evaporation conditions. Present-day monitoring studies depict low d-excess values for

subtropical moisture, contrasting with high d-excess values for moisture from sea ice margin areas (e.g. Steen-Larsen et al, 2015; Pfahl et al, 2014). The d-excess is also affected by distillation, and will decrease in polar regions if $\delta^{18}O_{ice}$ increases without any moisture source change. As noted above, available LIG d-excess data (Table 1) are slightly (insignificantly) above from pre-industrial. A stable or higher d-excess level together with an $\delta^{18}O_{ice}$ increase is therefore understood to reflect a slight poleward shift of moisture sources. In turn, this

would imply reduced distillation for $\delta^{18}O_{ice}$ and a reduced slope of the relationship between $\delta^{18}O_{ice}$ and temperature (Sime et al, 2013) in line with the average Holocene calibration.

### 3.2 Reconstructions based on air $\delta^{15}$N

In the absence of abrupt surface temperature change, $\delta^{15}$N is only affected by the gravitational signal linked to firn LID; the latter is directly related to changes in temperature and accumulation rate. Thus, if accumulation is

known, past temperature changes can be inferred from $\delta^{15}$N. As neither accumulation nor temperature is independently known for the LIG, we have to constrain the accumulation/temperature relationship in the past based on observation and/or models. We now describe the different steps of our procedure to estimate the temperature of the NEEM deposition site during the LIG from $\delta^{15}$N measurements.

### 3.2.1    Different estimates of the link between temperature and accumulation rate in Greenland

i: Accumulation rate and temperature can be linked through thermodynamic laws and ice sheet topography, despite significant uncertainties associated with atmospheric transport characteristics that lead to regional variability (Kaspner et al., 1995; Merz et al., 2014b). In a first approximation, temperature and moisture content of an air mass are linked through saturation pressure ("$P_{sat}$-approach"). This first order relationship between accumulation rate and temperature has long been used for Antarctic ice core chronologies (Lorius et al., 1985;

Ritz, 1992) with:





$$A(t) = A(t_0) \times \frac{\left[(\partial P_{sat}/(T+273))/\partial T\right]_t}{\left[(\partial P_{sat}/(T+273))/\partial T\right]_{t_0}} \qquad (2)$$

where $A(t)$ and $A(t_0)$ are the accumulation rates at time $t$ and $t_0$ respectively, $P_{sat}$ the saturation pressure over ice and $T$ the temperature in °C.

ii: Empirical relationships between accumulation rate and Greenland temperature have been provided by

different methods. The dataset obtained in 1952-1955 by Benson (1962) remains a reference today for evaluating surface accumulation rate reconstructions above the Greenland ice sheet (e.g. Hawley et al., 2014; Munk et al., 2003). Surface accumulation rate and temperature data from 146 sites show an exponential increase of accumulation rate versus temperature ("Benson-approach"). Within its associated 1σ envelope, it encompasses the accumulation rate versus temperature increase deduced from the "$P_{sat}$-approach" (Figure 3).

iii: More recently, Buchardt et al. (2012) used an array of 52 shallow ice cores spanning the last decades to centuries with accumulation rate estimates from annual layer counting on $\delta^{18}O_{ice}$ profiles. They identified different temperature vs accumulation rate relationships from one region to another. In the area of NEEM, NGRIP, GRIP and GISP2 deep ice cores, the "Buchardt-approach" leads to the following relationship in Greenland between accumulation, $A$ and temperature, $T$:

$$6.7 < \frac{1}{A} \times \frac{dA}{dT}(\%.K^{-1}) < 9.6 \qquad (3)$$

The results of the "Buchardt-approach" lie within the 1σ envelope of the "Benson-approach" (Figure 3). For the Northwestern Greenland area, in which the NEEM and NGRIP sites are located, the lower end of the accumulation rate/temperature sensitivity in equation (2) applies (Buchardt et al., 2012). The lower line of the Buchardt approach in Figure 3 is hence the most likely accumulation/temperature sensitivity for the NEEM ice

core. Note that one limitation needs to be stressed for the Buchardt and Benson approaches: the application of a spatial accumulation/temperature relationship makes the assumption that this sensitivity can be also applied to a temporal change, an assumption that, similar to the case of the spatial $\delta^{18}O$/temperature gradient, must not necessarily hold back in time.

iv: Masson-Delmotte et al. (2015) used estimates of snow accumulation rate and ice $\delta^{18}O$ in four shallow cores

in the NEEM area together with accumulation rate, temperature and when possible snowfall $\delta^{18}O$ reconstructions from different models simulation (ECHAM5, Global Climate Model developed by the Max Planck Institute for Meteorology; LMDZ, Global Climate Model developed by the Laboratoire de Météorologie Dynamique; MAR, Regional Atmosphere Model) are nudged to available climate reanalyses over the 1979 – 2007 period. In addition to model outputs or temperature gridded reconstruction (Box et al., 2009), the

amplitude of temperature increase at NEEM can also be estimated using borehole temperature measurements. Gathering the different sources of information for the strong warming period of 1979-2007 leads to a temporal slope between accumulation rate and temperature of 11 ± 3 %.°C$^{-1}$. This "Masson-Delmotte approach" gives much higher slopes than those obtained by the "Buchardt approach" (Figure 3). These high slopes are mainly linked to atmospheric model outputs (for temperature or accumulation), and we regard them as an upper limit.

By reducing the data used to infer the accumulation rate versus temperature slope to observations (NEEM accumulation rate deduced from shallow cores, borehole profile inversion, gridded reconstruction or





instrumental temperature measurements from the Greenland southwestern coast), the slope is only of $9 \pm 1\%.°C^{-1}$ which, in this case, is consistent with the "Buchardt approach".

v: Alternative estimates of accumulation at the NEEM deposition site are also provided by $^{10}Be$ data. Sturevik-Storm et al. (2014) compiled mean Holocene $^{10}Be$ concentration over eight Greenland sites and determined the

spatial relationship against mean accumulation rate estimates. This relationship was then applied to LIG $^{10}Be$ data from NEEM, leading to the conclusion that accumulation rate was 65-90% higher than today at the NEEM deposition site ("$^{10}Be$-approach"). However, the present-day spatial gradients in $^{10}Be$ concentration are caused by spatially varying contributions of wet deposition to the overall $^{10}Be$ deposition, assuming a homogeneous atmospheric $^{10}Be$ aerosol deposition over Greenland. This latter assumption implies that the atmosphere above

Greenland is well mixed with respect to $^{10}Be$ after transport from the location of troposphere/stratosphere foldings. The latter are the main entrance pathways of stratospheric aerosols into the extratropical northern hemisphere troposphere. The LIG climate is characterized not only by enhanced precipitation above Greenland (accumulation) but also during aerosol transport to Greenland. As a result, higher scavenging of $^{10}Be$ bearing aerosol en route must have led also to a lower atmospheric $^{10}Be$ concentration over Greenland than today. The

LIG accumulation estimate by Sturevik-Storm et al. (2014) is therefore likely an overestimation. More generally, the use of other chemical aerosol species as accumulation rate tracers is hampered by potential changes in the LIG atmospheric concentrations due to emission changes. Qualitatively, a correction of deposition effects using the lower end of Buchardt et al. (2012) approach (representative for Northwest Greenland), leads to LIG atmospheric concentrations of all chemical aerosol tracers similar to today. In contrast

much higher LIG accumulation rates as estimated by Sturevik-Storm et al. (2014) or no changes in accumulation between the LIG and the Holocene imply an unrealistic decrease in atmospheric aerosol concentrations for several aerosol tracers. Based on the changes in various chemical tracers in the ice (sea salt aerosol, biogenic aerosol, mineral dust) we conclude that the LIG accumulation must have been at least 20 % higher than pre-industrial and was likely similar to the Buchardt approach for Northwest Greenland.

vi: atmospheric general circulation model outputs do not suggest important changes in accumulation rate for the LIG compared to pre-industrial values in line with the small temperature changes of mean surface temperature (section 3.3). An informal comparison of some of the models (presented in Lunt et al., 2013) shows very limited accumulation increase over central Greenland, only up to 10 cm w.e.yr$^{-1}$, which corresponds to an increase of less than 5%. Stronger increases in accumulation rate at LIG associated with significantly warmer than pre-

industrial temperature were obtained in relation with a reduction of sea ice in the Nordic Seas (10% increase in accumulation, Merz et al., 2016). Finally, it has been shown that the geometry of the Greenland ice sheet and topographic changes can lead to various local accumulation scenarios for the LIG at the upstream NEEM deposition site (Merz et al., 2014b): depending on the prescribed LIG ice sheet topography, the modelled accumulation rate at LIG can be 25% lower to 13% higher than the pre-industrial accumulation rate. The lowest

estimate is linked to a change in the trajectory of air mass to the NEEM deposition site, with an increased eastward origin. Such a scenario can be assessed by evaluating d-excess variations. The data presently available shows similar d-excess levels at the LIG and pre-industrial periods, and therefore does not support a significant change in moisture source and trajectory (Table 1).





Often associated with relatively small temperature changes, these modelled accumulation rate scenarios for the LIG are in general lower than the accumulation rate scenarios discussed above and in contradiction with the recent modelled accumulation evolution depicted by Masson-Delmotte et al. (2015).

From the limitations of each accumulation rate estimates highlighted above, we conclude that the subset of relationships with the highest accumulation rate - temperature sensitivity may overestimate the accumulation response, and are less reliable. Accordingly, we argue the saturation pressure and the lower end of the Buchardt approach to be more likely to bracket the true LIG accumulation rate increase.

### 3.2.2 Measured versus modeled evolution of $\delta^{15}N$ with respect to temperature and accumulation rate changes

Our $\delta^{15}N$ data is compared with those simulated using a firn densification model forced by these different accumulation rate versus temperature relationships for the LIG. The firnification model relates LID to accumulation rate and temperature. Here, we use the Goujon et al. (2003) model in steady state to calculate LID, and the barometric equation (1) to translate LID changes into $\delta^{15}N$ changes.

The model correctly captures the present-day $\delta^{15}N$ values for NEEM and NGRIP, using the current mean values 15 for accumulation rate and temperature (Figure 3). At NEEM and NGRIP, firn studies have recently provided an accurate determination of the LID and $\delta^{15}N$ profiles (Guillevic et al., 2013; Buizert et al., 2013). At GISP2, there is no proper determination of the LID due to discontinuous sampling of air bottles and a large scatter of $\delta^{15}N$ values measured at the bottom of the firn, ranging between 0.305 and 0.325 ‰ (Bender et al., 2006). An average value of 0.31‰ is obtained from high-resolution $\delta^{15}N$ measurements over the last 4000 yr on the GISP2 20 core (Kobashi et al., 2008). For present-day, our simulation at GISP2 ($\delta^{15}N$ of 0.325‰) therefore lies at the upper limit of available measurements (Table 1).

### 3.2.3 Reconstructing Greenland LIG temperature

In order to estimate the LIG firn temperature (at the deposition sites), Figure 3 displays the $\delta^{15}N$ data points for each ice core site on contours of the simulated $\delta^{15}N$ values as a function of temperature and accumulation.

For the NEEM LIG deposition site, we detail below the graphical determination of the temperature of the firn column. The intersection between the $\delta^{15}N$ contour and the "accumulation rate vs temperature" evolution curve gives the range of realistic LIG accumulation rate (y-axis) and temperature (x-axis). Note that the "Benson-approach" is scattered, due to local surface topography effects, and therefore not directly relevant for the NEEM LIG temperature reconstruction. We have used this "Benson-approach" only for validation of the other 30 accumulation rate vs temperature relationships.

Let us assume that the LIG accumulation rate at the NEEM deposition site was the same as today at NEEM, despite increasing temperature. In this conservative but unrealistic case, our $\delta^{15}N$ data point to a 3.5°C warmer firn column. Assuming a minimum 20% accumulation increase at the NEEM deposition site leads to an estimate of 4.5°C surface warming between the NEEM upstream deposition site and the current NEEM firn temperature.

From the intersection between the $\delta^{15}N$ level measured in the NEEM LIG section and the lower accumulation rate estimates ("$P_{sat}$-approach" and "Buchardt approach), we obtain a larger estimate of 6-7°C warming of the



firn column at the NEEM LIG deposition site compared to the current NEEM firn temperature. This corresponds to an accumulation rate of 26-30 cm water equivalent $yr^{-1}$, i.e. 32-50% higher than the present-day accumulation rate at NEEM. The highest LIG warming compatible with the $\delta^{15}N$ data (-20°C, i.e. about 9°C above present-day NEEM values at the upstream NEEM deposition site) corresponds to an accumulation rate of 46 cm water

equivalent.$yr^{-1}$, i.e. 130% higher than the present-day accumulation rate at NEEM, using the highest slope for the relationship between accumulation rate vs temperature (highest limit of "Buchardt approach" and "Masson-Delmotte approach"). We argued earlier that this latter high-end approach overestimates the true Eemian accumulation rate. For the $^{10}Be$ approach, which also represents an upper limit of the possible accumulation increase we find a LIG temperature 7-8°C warmer than at the current drilling site (without ice sheet altitude

correction).

At NGRIP, the same graphical approach leads to an estimated temperature of -28.3 ± 0.7°C at the end of the LIG (120 ka on the AICC2012 timescale) compared to -31.5°C for pre-industrial, i.e. a difference of +3.2°C between LIG and pre-industrial. Even if NGRIP is not on a dome, the upstream effect is quite small: the NGRIP LIG deposition site is estimated to lie 48 km upstream in the direction of Summit with small associated altitude

gradients between NGRIP and Summit (Buchardt, 2012). The LIG uncertainty range arises from the uncertainty in the accumulation rate vs temperature relationship. This implies warming by 3.2± 0.7°C at 120 ka, with a 14-37% higher LIG accumulation at NGRIP. This is likely an underestimation of the full warming range encompassed during the LIG, because the NGRIP ice core does not extend towards the warmest part of the LIG. At the NEEM deposition site however, the temperature estimated following the graphical method of Figure 3

from the $\delta^{15}N$ value at 120 ka (0.256 ‰) is only 0.5°C lower than the estimated LIG optimum temperature, hence a difference of +8.5 ± 2°C between LIG and pre-industrial.

At Summit, only very high temperature can be reconciled with LIG $\delta^{15}N$ values 0.7 to 0.9‰ lower than today: -20°C to -16°C according to graphical determination using Figure 3 compared to a pre-industrial temperature of -31.7°C. Still, without a continuous sequence of interglacial ice at Summit, the true origin of the $\delta^{15}N$ signal at

Summit is doubtful. We cannot yet assess whether this signal is purely gravitational or whether it is dominated by a thermal signal or a firn hiatus effect of ~16 m as estimated from the barometric equation (1).

Finally, note that the present-day values in accumulation, temperature and $\delta^{15}N$ at NGRIP and NEEM nicely align with the lowest accumulation-temperature relationships (Buchardt and $P_{sat}$ approaches; Figure 3), while the highest accumulation rate/temperature sensitivities are not consistent with a common sensitivity at NEEM

and NGRIP. At Summit, the current accumulation rate is significantly higher than expected from the Northwest Greenland Buchardt estimate, indicating that other advective moisture pathways come into play, consistent with analyses of spatial influences of weather regimes in Greenland (Ortega et al, 2014).

### 3.2.4   Limitations of the $\delta^{15}N$ based temperature reconstruction at the NEEM sites

In the following, limits inherent to this $\delta^{15}N$ approach are highlighted, which shall motivate further studies to

refine the temperature estimate. First, we have applied a firnification model optimized for present-day central Greenland firn to past periods with different, warmer conditions, outside the range of model validation. For instance, the occurrence of substantial summer melt could accelerate firn densification and produce a smaller





close-off depth (and therefore smaller $\delta^{15}$N values) than expected from the Goujon model for a given temperature. In principle, the validity of firn models in such temperature range can be tested, if firn studies are performed in Greenland sites which are today warmer than the central deep drilling sites of Summit, NGRIP and NEEM, but no data are yet available.

Second, the relative changes in accumulation rates and temperature between the NEEM deposition site and NEEM remain difficult to estimate. While we have gathered all the available information on the spatial and temporal relationships between surface temperature and accumulation rates, other influences need to be ideally considered, such as changes in regional atmospheric circulation associated with a different climate and modifications in ice sheet topography (Merz et al., 2014a;b) and/or sea ice extent (Merz et al., 2016).

Finally, we have identified a sharp signal in the NEEM LIG $\delta^{15}$N profile, at 2384 m (121 ka), with no parallel signal in $\delta^{18}$O$_{ice}$ or chemical records. This signal challenges our attribution of $\delta^{15}$N variations solely to changes in accumulation rate and/or temperature, and suggests potential influence of surface melt on firn depth, LID and therefore $\delta^{15}$N. While the overall stability of the NEEM $\delta^{15}$N record over the LIG supports a gravitational / climatic interpretation, a dominant influence of surface melt of about 16 m cannot be excluded for the Summit

ice core sections associated with very low $\delta^{15}$N (0.23 ‰).

### 3.3    LIG temperatures in Greenland as estimated by climate models

The LIG climate has been simulated by a suite of climate models of various complexities. Most of these simulations are included in the model intercomparison studies of Lunt et al. (2013) and Bakker et al. (2013, 2014). The former study compared equilibrium ("snap-shot") simulations, covering time-slices within the early

LIG (125-130 ka), to temperature proxy data, whereas the latter studies discussed transient simulations covering the entire LIG. The model mean of all equilibrium simulations computes an annual mean temperature increase over Greenland between 0 and 2°C with respect to pre-industrial control simulations (Fig. 6a of Lunt et al., 2013). At the upstream NEEM depositional site the same range of annual mean temperature increase is found when analysing the individual model simulations for the 125 ka time slice (0.2-2.2°C, Figure 4). The majority of

the transient models simulate a maximum early LIG temperature increase of similar magnitude as the equilibrium model simulations. Two exceptions are the MPI-UW model that computes an annual mean temperature increase as high as ~4°C at around 126 ka. While the other exception is the transient simulation of CCSM3, which consistently simulates lower annual mean temperatures for the LIG compared to pre-industrial.

The LIG temperature and precipitation patterns produced by climate models can be applied as a forcing to ice

sheet models simulating the LIG evolution of the Greenland ice sheet (e.g. Otto-Bliesner et al., 2006; Robinson et al., 2011, Born and Nisancioglu, 2012; Stone et al. 2013; Langebroek and Nisancioglu, under review). Several uncertain parameters within these ice sheet models need to be considered (e.g. basal sliding parameter). This, in combination with uncertainties in the schemes translating the large-scale climate forcing to the local Greenland mass balance schemes, results in a large ensemble of possible melting scenarios for the Greenland ice sheet for

each individual ice sheet model study. Present-day observations and paleo proxy data help to reduce this large spread. In particular, the paleo information of limited surface elevation reduction at the central ice core locations strongly constrains the simulated LIG Greenland ice sheet evolution (Masson-Delmotte, 2013).





Here, we investigate the annual mean surface temperature anomaly at the upstream depositional site of NEEM for simulations fulfilling the (paleo) data constraints in the ice sheet modelling studies of Stone et al. (2013) and Langebroek and Nisancioglu (under review). For these simulations, the corresponding annual mean surface temperature anomaly at the upstream NEEM depositional site is around 1-4°C above pre-industrial. The annual

mean LIG temperature forcing applied in these two studies is similar (Fig. 4, HadCM3_Bris vs. NorESM_BCCR), whereas the summer temperature anomaly is about 1°C larger over central Greenland in Stone et al. (2013). Also, the ice sheet models and the methods for calculating the surface mass balance from the climate forcing are different. However, the surface temperature anomaly scenario in these studies is mainly restricted by the limited surface elevation reduction during the LIG implied by NEEM data. Ice sheet

simulations with a higher surface temperature anomaly than approximately 4-5°C indicate a too large elevation lowering at the ice core locations, and are therefore rejected.

In these studies, the feedback of changes in surface elevations on the climate is either not included (surface topography is kept fixed, e.g. Langebroek and Nisancioglu, under review) or not well resolved due to coarse model resolution (Stone et al., 2013). However, Merz et al. (2014a) showed that a steeper surface slope can

cause an additional 1-3°C SAT increase due to an increase in katabatic winds which foster downward flux of sensible heat. In order for this effect to be important for NEEM, the ice sheet geometry needs to change such that its LIG depositional site is closer to the rim of the ice sheet, e.g. with large melting of northeast Greenland. Confirming the suggestion of Sime et al (2013), Merz et al. (2016) used the CCSM3 and CCSM4 models to demonstrate that the large spread in Greenland SAT change among the LIG equilibrium simulations (Figure 4,

Lunt et al. (2013)) is mostly due to differences in simulated sea-ice extent. They showed that SAT and accumulation changes at the NEEM LIG deposition site are particularly sensitive to sea-ice retreat in the Nordic Seas.

In summary, studies performed with atmospheric models suggest that a number of processes may combine to produce larger amplitude of warming as simulated in state-of-the-art coupled climate models. They suggest that

Greenland LIG SAT change can be amplified in response to regional sea ice retreat and in response to change in the ice sheet topography. For the NEEM deposition site, LIG annual mean surface temperatures of approximately 5°C above pre-industrial can be obtained: 2°C being due to LIG external forcing (orbital and greenhouse gases), 0.6 to 2.3°C to be attributed to a decrease of sea ice in the Nordic Seas and 1 to 2 °C associated with a moderately smaller GrIS. Higher temperatures might be possible in more extreme scenarios,

however then it is unlikely that the ice sheet can maintain surface elevations relatively close to modern for all central deep ice core locations as suggested by the ice core records. The coherency between scenarios of large SAT warming and the plausability of the Greenland ice sheet response still remains to be fully explored. It includes specific analyses of the seasonal aspects of the SAT change

## 4. Conclusions and perspectives

In this study, we have compiled the Greenland ice core data and methods available to quantify LIG temperature change. New estimates of temperature change were provided based on $\delta^{15}N$ and firn densification modeling, independent from water stable isotope data. They imply that the mean annual firn temperature at the LIG deposition site, upstream of the current NEEM site, experienced 7-11°C warming, without accounting for





changes in elevation related to ice thickness change. As a comparison, the initial estimate of NEEM community members (2013) based on $\delta^{18}O$-temperature relationships calibrated on Holocene data led to a LIG surface air temperature at the upstream NEEM deposition site 5.7-9.3°C warmer than at pre-industrial (NEEM comm. members, 2013). The approach based on water stable isotopes strongly depends on the calibration of the isotope-temperature relationship. NEEM (2013) used Holocene calibrations based on central Greenland ice cores (0.5 ‰.°C$^{-1}$). However, Masson-Delmotte et al (2015) showed that water stable isotopes were twice more sensitive to temperature changes at NEEM during recent decades than inferred for the Holocene in central Greenland. Alternatively, we can independently quantify the isotope-temperature relationship using our new data: the $\delta^{15}N$ temperature reconstruction of a 4.5 to 8.5°C temperature change for a 2.1‰ change between NEEM pre-industrial and the NEEM deposition site at LIG (hence 2 different sites at 2 different periods) leads to a slope of the isotope-temperature slope (0.25 to 0.46 ‰.°C$^{-1}$), even smaller than the result of the central Greenland Holocene calibration, and comparable to the LGM vs Holocene temporal slope in Greenland (0.3 ‰.°C$^{-1}$) (Cuffey et al., 1995; Dahl-Jensen et al., 1998). This finding implies that the isotope-temperature relationship inferred for recent warming (associated with e.g. sea-ice retreat in the Baffin Bay area) does not apply for warmer than today LIG conditions. Such a low slope is also captured in some isotopic enabled atmospheric models for LIG conditions (Masson-Delmotte et al, 2011), but results appear dependent on patterns of sea-surface-temperature changes and associated moisture origin changes (Sime et al, 2013). Indirectly, our constraints on the isotope-temperature relationships can therefore help to test the realism of large-scale climate patterns (including atmospheric transport response to sea surface temperature and sea ice changes) simulated in response to orbital forcing, beyond just Greenland temperature.

In addition to the $\delta^{15}N$ temperature reconstruction, ice core data provide multiple lines of evidence of significantly warmer conditions, at least during summer, at the upstream NEEM deposition site during the LIG than today at the NEEM site: in-situ $CH_4$ production likely due to summer melt, firn shrinking suggested by the $\delta^{15}N$ peak at 2384 m. The evidence of summer melt in the LIG section of the NEEM core also stresses the intrinsic limitations of our approach. Melting-refreezing, which accelerates firnification processes, is not included in our firnification model, which, strictly speaking, is only valid for the dry snow zone. Still, the occurrence of extensive summer melt implies that mean summer temperatures at the site of deposition frequently reached the melting point, which is about 5°C higher than current mean summer temperatures at NEEM. The puzzling disappearance of ice from the penultimate deglaciation at the bottom of the Greenland ice cores may result from such warm conditions affecting melt and/or ice flow.

Large warming at the NEEM deposition site is difficult to reconcile with climate simulations: in response to orbital forcing and greenhouse gases concentration forcing only, state-of-the-art intermediate complexity or fully coupled climate models mostly produce an annual mean temperature increase of less than 2°C above pre-industrial present-day during the LIG in NW Greenland. However, sea ice cover retreat in the Nordic Seas and changes in Greenland ice sheet topography may significantly enhance surface warming (Merz et al., 2014a; 2016) and therefore reduce the gap with our estimate. However, annual mean warming by 5°C or more leads to simulations of larger ice sheet retreat than implied by data constraints on the LIG ice sheet elevation: the NEEM paradox still needs to be solved, possibly through a stronger focus on summer rather than annual mean temperature change.





Further work is required to overcome the unavoidable limitations of firn our temperature estimate. New firn monitoring studies in Greenland areas affected by summer melt and in today's ablation zone are crucially needed to improve firn modeling and interpretation of the $\delta^{15}N$ signal, especially for the sharp anomaly suggested to reflect 15 m firn shrinking. Similarly, monitoring of water stable isotopes in the Arctic water vapor

is also critical to better understand and model the relationships between atmospheric circulation, moisture transport pathways, snow-vapour isotopic exchanges and the isotopic composition above the Greenland ice sheet.

### Acknowledgments

NEEM is directed and organized by the Centre of Ice and Climate at the Niels Bohr Institute and US NSF,
Office of Polar Programs. It is supported by funding agencies and institutions in Belgium (FNRS-CFB and FWO), Canada (NRCan/GSC), China (CAS), Denmark (FIST), France (IPEV, CNRS/INSU, CEA and ANR), Germany (AWI), Iceland (RannIs), Japan (NIPR), South Korea (KOPRI), The Netherlands (NWO/ALW), Sweden (VR), Switzerland,, the United Kingdom (NERC) and the USA (USNSF, Office of Polar Programs) and the EU Seventh Framework programs. The Division for Climate and Environmental Physics, Physics Institute,
University of Bern, acknowledges continuing support by the Swiss National Science Foundation as well as access to the infrastructure at the Swiss National Supercomputing Centre. The HadCM3 climate model simulations were carried out using the computational facilities of the Advanced Computing Research Centre, University of Bristol - http://www.bris.ac.uk/acrc/. This project benefitted from support from Past4Future 'Climate change - Learning from the past climate', a Collaborative Project under the 7th Framework Programme
of the European Commission (grant agreement no. 243908). P.M.L. is funded by the Research Council of Norway through the IceBed project (221598). This work was supported by European Research Council Advanced Grant no. 246815 WATERundertheICE. The research leading to these results has received funding from the European Research Council under the European Union's Seventh Framework Programme (FP7/2007-2013) / ERC grant agreement n°306045 (COMBINISO).

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





| | Pre-industrial[*] | | | | | LIG[**] | | | |
|---|---|---|---|---|---|---|---|---|---|
| | NEEM | NEEM deposition site | NGRIP | GRIP | GISP2 | NEEM deposition site | NGRIP | GRIP | GISP2 |
| $\delta^{15}N$ (‰) Uncertainty : ± 0.007‰ | 0.290[1] | | 0.310[2] | | 0.305-0.325[3] | 0.251 | 0.290 | 0.230 | 0.230 |
| $\delta^{18}O_{ice}$ (‰) Uncertainty : ± 0.1‰ | -33.6[4] | -35[7] | -35.5[4] | -35.2[4] | | -31.5[7] | -32.2 | -31.8 | -31.8 |
| d-excess (‰) Uncertainty : ± 1.5‰ | 11[4] | 11 | 10.5[4] | 9.5[4] | | 11.5 | 9.9[5] | 10.8[6] | |
| Mean temperature (°C) | -28.5[8] | -31[7] | -31.5[2] | -31.7[4] | | | | | |
| accumulation (m i.e. yr$^{-1}$) | 0.22[8] | | 0.19[2] | 0.23[2] | | | | | |

**Table 1: Characteristics of Greenland deep ice core at pre-industrial and during the LIG.**

The uncertainty corresponds to the standard error of the mean. Where data was compiled from previous studies, the references for the number in this table are taken from: 1-Buizert et al. (2012); 2-Guillevic et al. (2013); 3-Kobashi et al. (2008) - Bender et al. (2006); 4- Masson-Delmotte et al. (2005); 5- Capron et al. (2012); 6-Jouzel et al. (2007) - 7- NEEM comm. members (2013)- 8 –Masson-Delmotte et al. (2015)

[*]: The accumulation rate, $\delta^{18}O$ and d-excess values attributed to the pre-industrial conditions correspond to averages over the last 200 years. For $\delta^{15}N$, it corresponds to the value at the bottom of the firn (itself built over the last 200 years). The mean temperature is issued from borehole measurements.

[**]: For determining the $\delta^{18}O_{ice}$, $\delta^{15}N$ and d-excess values attributed to the LIG, we have taken the average of the corresponding records for NEEM, GRIP and GISP2 between 122 and 126 ka (excluding negative $\delta^{15}N$ peak at 2384 m at NEEM). At NGRIP, we probably miss the first part of the LIG and probably the optimum of the LIG. The values indicated here correspond to the deepest level of the NGRIP ice core dated at 120 ka on the AICC2012 timescale.





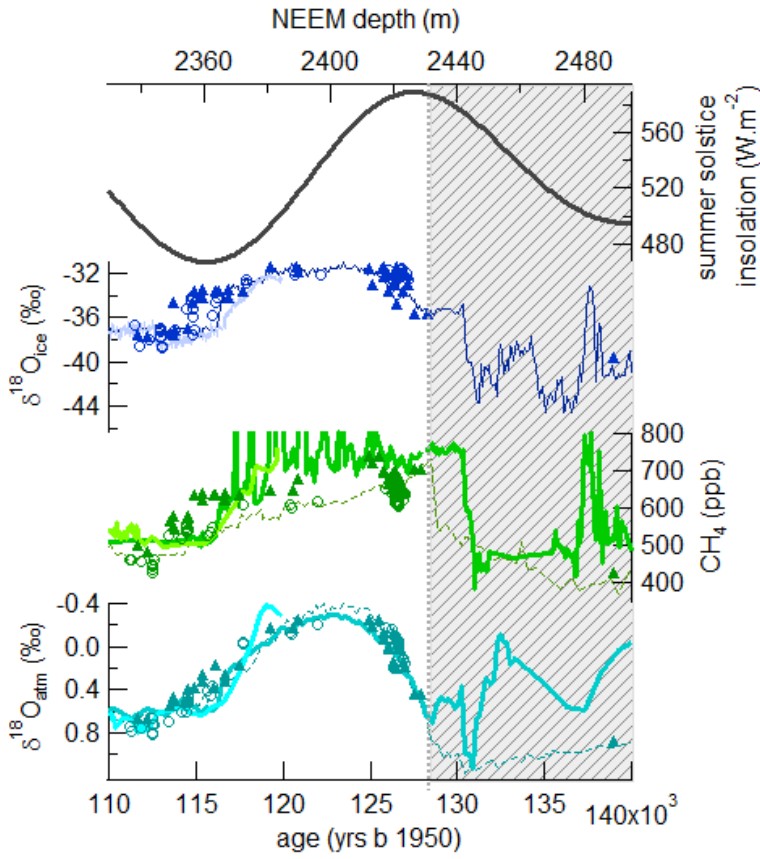

**Figure 1: Synchronized Greenland $\delta^{18}O_{ice}$ records.**

From top to bottom: summer solstice insolation at 77°N (black); $\delta^{18}O_{ice}$ from NGRIP (light blue line), GRIP

5      (open circles) and GISP2 (triangles) on the AICC2012 timescale (bottom axis) and NEEM $\delta^{18}O_{ice}$ (dark blue

line) on its depth scale (top axis); $CH_4$ records from NGRIP (light green line), GRIP (open circles), GISP2

(triangles) and EDC (dashed line) on the AICC2012 timescale and NEEM (dark green line) on its depth axis ;

$\delta^{18}O_{atm}$ records from NGRIP (light blue), GRIP (open circles), GISP2 (triangles) and EDC (dashed line) on the

AICC2012 timescale and NEEM $\delta^{18}O_{atm}$ (turquoise) on its depth axis. The shaded grey rectangle highlights the

10     deepest part of the NEEM records, where no gas synchronization with Antarctic ice core records from the

penultimate glaciation is feasible.



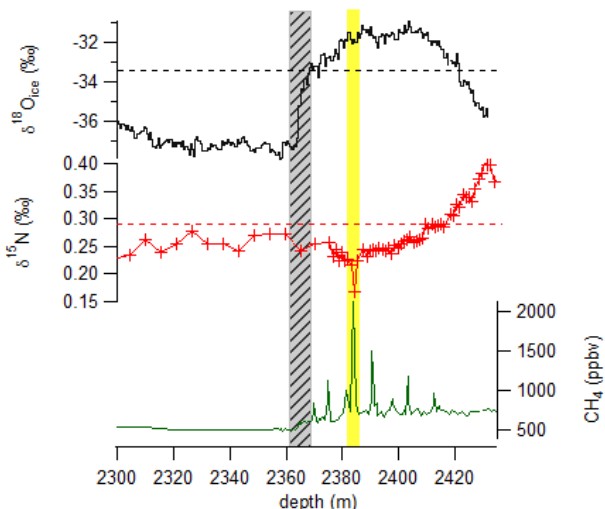

**Figure 2: NEEM $\delta^{18}O_{ice}$ (top), $\delta^{15}N$ (middle) and $CH_4$ (bottom) spanning the LIG.**

The grey rectangle indicates the stratigraphic disturbance identified at the upper part of the LIG section. The yellow rectangle highlights the single negative peak in $\delta^{15}N$ at NEEM during the LIG, corresponding to the strongest positive peak in $CH_4$, both peaks being identified at 2384 m depth.




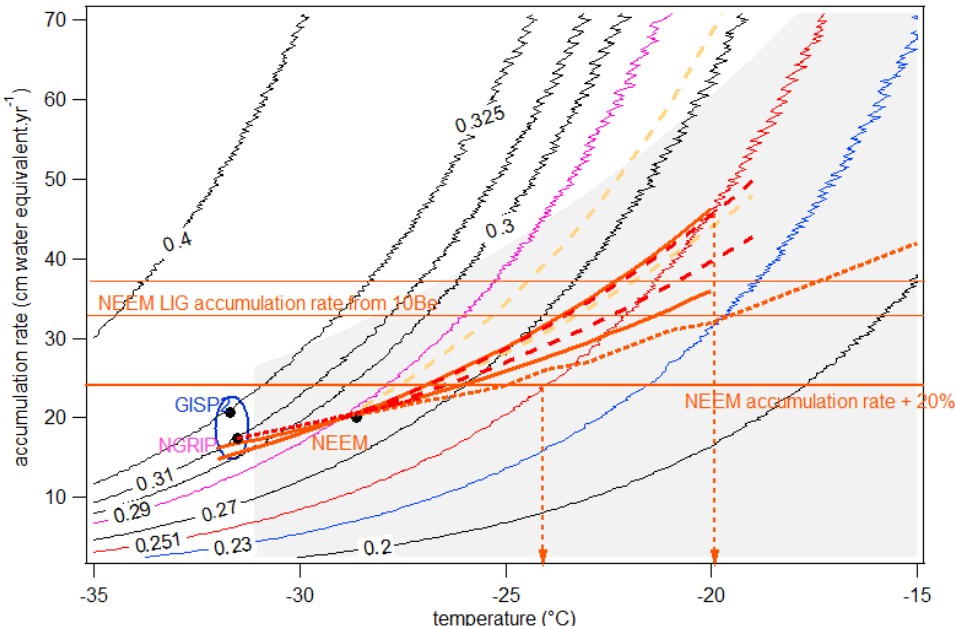

**Figure 3: Contour plot showing the evolution of modeled $\delta^{15}N$ with respect to accumulation rate and temperature (Goujon et al., 2003).**

5   The black circles indicate the accumulation rate vs temperature for pre-industrial conditions for the 3 Greenland sites discussed here, Summit, NGRIP and NEEM. The grey zone corresponds to the repartition of the accumulation rate vs temperature data with a $1\sigma$ envelop from Benson et al. (1962). The dotted red line (i), the bold solid lines (ii), the dashed light orange lines (iii), the dashed bold red lines (iv) and the thin horizontal red lines (v) indicates the relationship between accumulation and temperature deduced respectively from the "Psat-
10   approach" (i), the "Buchardt-approach" (ii), the "Masson-Delmotte approach" (iii), the "Masson-Delmotte approach" using only instrumental data (iv), the "$^{10}Be$ approach" and a minimum estimate from other chemical compounds in the ice "chemistry approach" (v).   The red contour line indicates the NEEM LIG $\delta^{15}N$ level excluding the peak at 2384 m. The two vertical arrows indicate the highest and lowest NEEM LIG temperature as determined by the graphical determination (see text). The thin pink contour line indicates the NGRIP LIG
15   $\delta^{15}N$ level and the blue contour line indicates the GRIP/GISP2 LIG $\delta^{15}N$ level. The blue circle stands for the possible present-day values for GISP2 $\delta^{15}N$.



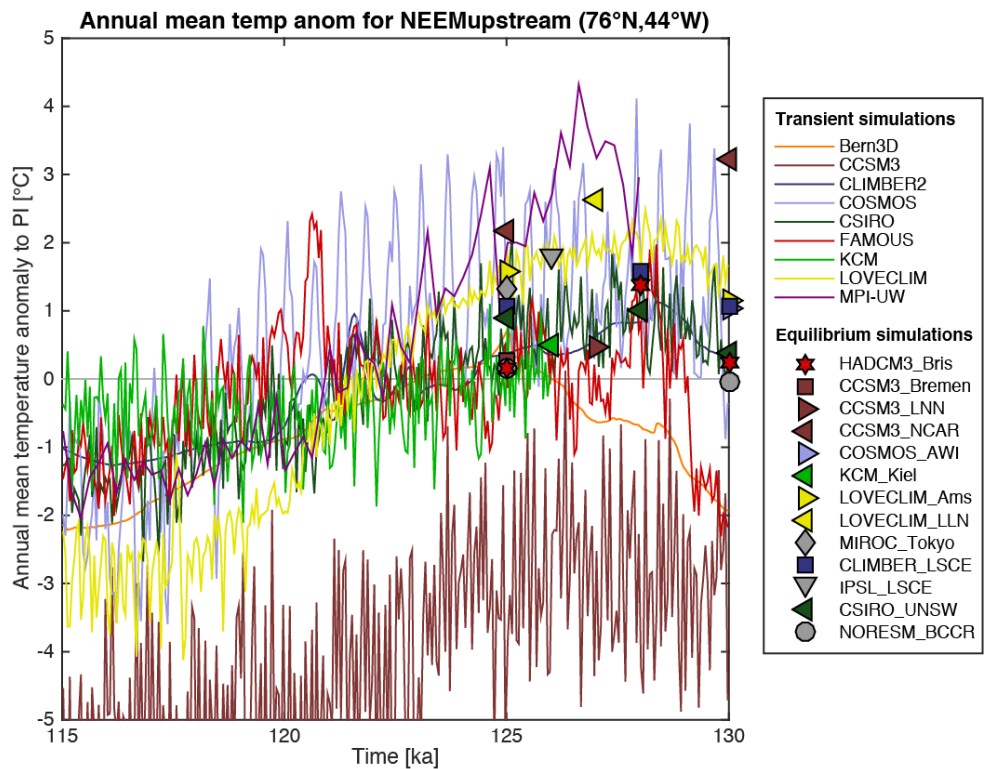

**Figure 4: Annual mean temperature anomaly relative to pre-industrial control simulations for the upstream NEEM depositional site extracted from transient and equilibrium climate model simulations.**

Note the similarity for the annual mean temperature increase at 125 and 130ka as simulated by HadCM3_Bris and NorESM_BCCR. For more information on the climate models and the simulations themselves we refer to the compilation studies and references therein (Bakker et al., 2013, 2014 and Lunt et al., 2013).

