# Peer review of "How warm was Greenland during the last interglacial period?"

_Climate of the Past, 2016_

## Short Comment (SC1) · 7 Apr 2016

I find this paper very interesting and I hope to see it published. However, I have a few comments and ideas that I think would improve the quality of the paper and might also give it a slightly greater impact.

The last interglacial is clearly an example of when almost all climate models agree on the sign of the climate change temperature signal but generally underestimate its magnitude with respect to available proxy data. The discussion about this discrepancy is, however, very qualitative as it stands and it would be interesting to see an attempt to bridge the gap between the data and modeling communities. The most obvious way to do that is to improve how model data is used in the analysis.

1: I am somewhat critical to how you use climate model data in the current version

of the paper. First, all models are to varying degree simplified versions of reality so a perfect match with proxy data is not to be expected. Secondly, comparing proxies with data in a single grid cell is arguably a misuse of climate models as they are designed to give an indication of the average conditions over a large region (the size of the region is dependent on the model complexity and grid resolution but a rule of thumb is to use at least a few grid cells). You should perhaps also comment on the range of models used in Fig. 4; the figure presents data from EMICS to full GCMs, which are worlds apart both in terms of complexity and modeling strategy (e.g. data constrained vs. free running, highly simplified vs. very complex, etc).

2: You mention that the $\delta^{18}$O signal recorded in ice cores can be influenced by changes in transport pathways and precipitation seasonality. The former is a bit tricky to investigate but you can easily perform a similar analysis as in Pausata and Löfverström (2015) (On the enigmatic similarity in Greenland $\delta$18O between the Oldest and Younger Dryas, Geophys. Res. Letters, 42, doi:10.1002/2015GL066042) and quantify the importance of precipitation seasonality and cloud temperature for the implied $\delta^{18}$O signal in the models.

3: I would be careful citing unpublished work or papers in open discussion, except of course if the papers are accepted and about to be released. There is never a guarantee that a paper will be accepted only because it is in review and the methodology and conclusions might change significantly when the paper is finally published.

---

## Referee Comment (RC1) · Anonymous Referee #1 · 26 Apr 2016

Review: "How warm was Greenland during the last interglacial period?" by Amaelle Landais et al.

The manuscript looks at d15N evidence, to help determine the peak LIG (Last Interglacial) temperature across Greenland. This is an important question, with implications for the resilience of the Greenland ice sheet to future temperature rise. The manuscript is mostly well written and structured. However, I have a few reservations about the way the d15N evidence is presented.

Major points:

p11 It is very interesting that the NEEM and NGRIP d15N reconstructions at 120 ka are so different: +8.5 versus +3.2C. This merits more analysis and attention. In particular, is the 120 ka NGRIP is less affected by uncertainty generated by melt processes?

This should be mentioned in the conclusions and in the abstract. Moreover the likely reasons for this discrepancy should be subject to further investigation/discussion and emphasis within this manuscript.

p12 L7-9 Have the studies by Metz et al not already provided enough information to attempt to constrain these 'other influences' on surface temperature and accumulation rates? And if so, should this not be included within the analysis/uncertainties for this study?

Minor points:

p6-7 section 3.1 This text could be clarified. It could be made easier for the reader to follow whether the reconstructions relate to the LIG or the Holocene.

p8 L22-23 "must not necessarily hold back in time" rephrase.

p9 L25 Capitalisation.

p11 L11-12 120ka is considerably later than the NEEM peak temperature, it could be made clear to the reader which different times in the LIG the reconstructions relate to, particularly the NEEM versus NGRIP.

p11 L31 Would be useful to have a brief description of the 'other advective pathways'.

p11 L42 "We hope?" shall motivate further studies?

p12 L1-4 I find this description of which further studies are required particularly useful.

p12 'sharp signal' rephrase?

p12 L14 16m? And perhaps clarify/rephrase 'a dominant influence of surface melt of about 16m'.

p12 L15- 37 This is a very nice summary review or relevant GCM studies.

P14 L24-25 "The evidence of summer melt at the LIG section of the NEEM core also stresses the intrinsic limitations of our approach." As above, this sentence/point could

be rewritten to emphasise the actual uncertainties associated with the d15N-based temperature estimates.

---

## Referee Comment (RC2) · EW Wolff (Referee) · 8 May 2016

The suggestion that Greenland could have been 8 degrees warmer in the last interglacial, and yet the ice sheet survived, is indeed a paradox. It is really important to find other ways to tackle this. A recent paper (Masson-Delmotte et al., 2015) involving some of the same authors as this one concluded that the temperature change may only have been about half that, illustrating that it is very hard to pin down the magnitude of the temperature change based on the water isotopes. It is therefore very important to find alternative ways to assess the temperature change that occurred. This paper takes a new approach, of using 15N data, with the assumption that these tell us the thickness of the firn at the time, and that this is controlled by temperature and snow accumulation rate. By making a range of assumptions about the accumulation rate (either as a function of temperature, or just as an independent adjustable), the authors

attempt to put limits on the possible temperature rise.

This is clearly an important problem and it is very worthwhile to try this approach. However, the authors reach a conclusion that I don't think their data justify, especially if one includes a further estimate of the accumulation rate, which I would rate as being at least as valid (perhaps more so) compared to the ones they choose. The critical part of the paper is Figure 3. I would make a first comment that, given how crucial this figure is, it's extremely hard to follow. I'll follow this up later. If I accept all the accumulation estimates shown (but I don't, see below), I reach the conclusion of the authors in section 3.2.3, but not the overall conclusion of the paper. This section seems to conclude:

For Summit, the data can probably not be interpreted as a pure thermal signal; For NEEM, you seem to choose a range between the +20% accumulation (4.5 degrees warming), and the "M-D" approach ($\sim$8 degrees warming). The authors don't explicitly state how they corrected for the upstream issue but I assume from Table 2 that they added 2.5 degrees. Thus the range they estimate is 7-10.5 degrees. For NGRIP, the best estimate is 3.2+/-0.7 degrees, with no significant upstream correction, and a possible extra 0.5 degrees for the warmest part of the LIG.

So taken at face value, one would conclude that one site gives 7-10.5 degrees and the other gives 2.5-4 degrees, which does not seem like a basis for asserting that the climate was 8 degrees warmer and that the NEEM paradox is confirmed.

However, what Figure 3 really tells us is that there is no good way of estimating the accumulation rate in the LIG, and therefore in the end we can't constrain the temperature this way. I can suggest another very valid way to estimate the accumulation rate. In Kapsner et al (1995), which the authors cite, an accumulation rate-temperature estimate for the early Holocene (pre-Boreal, which seems the most relevant) is made from correlation of annual layer thicknesses with oxygen isotope ratio measurements. They find a rather shallow slope within climate periods such as the pre-Boreal. Because they

actually did a regression of accumulation against oxygen isotopes, we can in fact estimate directly (without going through temperature) what should be the acc rate change for an oxygen isotope change of 3.5 permil (Table 2, change at deposition site). Given that they report a slope of 0.9%/K, and they used a calibration of 0.53 permil/K, their acc-isotope slope must have been 1.7%/permil, and a 3.6 permil change would correspond to a 6% change in accumulation rate. This would suggest a 3.5 degree warmer firn column, which adding the 2.5 degrees upstream correction, implies the NEEM region was 6 degrees warmer than present. Of course there are many reasons why this estimate may also be wrong. However, as a temporal slope, it is probably the most evidence-based estimate of all those presented, and should quite definitely be part of the range that is considered. Taking it together with the 4 degree estimate for NGRIP (which would also be reduced by assuming only a 6% change in accumulation, presumably to about 3 degrees), it does not suggest that the 8 degree conclusion is most likely right, but rather that it is probably too high, with values of 3-6 degrees more likely, pending more definitive estimates of accumulation rate. I know that I am playing Devil's advocate for a low value here, because really the evidence about acc rate is weak, but anyway I do not think the current conclusion of the paper (that it reinforces the paradox) is sustainable.

In the rest of this review I will go through the paper in more detail, and comment some more on Figure 3. However my overall suggestion is that, taking into account another realistic way of calculating accumulation rate, and the NGRIP result, the conclusions and abstract should be much more balanced and should not claim to be confirming the very high estimate of NEEM Project Members.

The abstract needs recasting in light of the NGRIP result and the more considered estimate of the NEEM accumulation rate that I am proposing.

Page 3, line 5 "rules out stratigraphic disturbance within the segment" – add this because clearly there is disturbance in the core as a whole.

[Figure]

Page 4, line 13. Is it worth here pointing out that the relationship most likely depends on changes in sea ice among other factors?

Page 5, line 27. You don't actually say it but I assume there really is a visible ice layer at this depth? I am not sure I see exactly how an event of the sort you imagine would affect 15N – after all a significant shrinkage of the firn must affect significantly the porosity, so it is not really clear what the net effect on the ability of 15N to diffuse should be, but since this is not crucial to the paper, I think there is no need to go further.

Section 3.2. While I think it is reasonable to assume that, in Greenland, 15N data under interglacial conditions should conform to a firn model with specified temperature and accumulation rate, one should add the small caveat that, for central East Antarctic sites, it has been rather conclusively shown that the firn models don't work correctly.

Section 3.2.1. As I already suggested, you really only show in this section that we have no good basis for estimating the accumulation rate that applies to this warm interglacial. Here I just comment on some of the estimates you give: i) The Kapsner paper quite conclusively shows that the thermodynamic approach is not really applicable to a situation like that in Greenland, where most of the precipitation is related to cyclones and storm activity. It's fine to mention it as an option, but it is clearly flawed. When it comes to figure 3, I really don't understand what the grey shaded area is meant to represent, so if you leave it, it needs a better explanation. ii) It seems as if this Benson approach is actually an empirical spatial version of the thermodynamic approach? However, while I could almost justify this approach for single storms tracked (and their integrated effect over a year) spatially across Greenland, there seems no basis for translating that into the temporal domain, as you acknowledge on Page 8, line 22. Note also that Benson 1962 seems missing from the reference list. iii) Same applies for the Buchhardt approach, but since the blue lines are missing from the plot I cannot assess it. I understand why you prefer the sensitivity Buchardt derived for the NW region but don't understand why you give a range for Greenland of 6.7-9.6%/K, when Buchardt's table gives a value of only 1.5%/K (with an error bar encompassing zero) for

the nearby NE region, and even a negative slope for SW Greenland. I do not think you have summarised the Buchhardt outcome fairly; in my opinion it gives weak evidence for a 6.7%/K slope, but with an uncertainty that encompasses a zero slope. Note also that it is unnecessary to draw a curve to represent Buchardt; he actually derived an acc rate-del18O slope, so for a measured change in del18O, you can pinpoint precisely the range of the change in acc rate and draw them as horizontal lines . iv) I have not checked the MD approach specifically, but note that this is the same paper that derived a much larger del18O-temperature slope and therefore a smaller temperature change than NEEM Project Members, so it seems inconsistent to use the same model simulations to derive the opposite result. v) I have not checked the 10Be estimates, but note only that because deposition in Greenland under interglacial conditions is completely dominated by wet deposition, it seems very unlikely that one can derive accumulation rates from 10Be. I agree with your implication that the change in overall scavenging would have to be accounted for in a sophisticated model, and I don't believe that local accumulation rate would be the dominant control on 10Be concentration or flux. For the same reason, I do not understand at all your statement on Page 9, lines 22-24 that changes in aerosol rule out no change in accumulation: if you want to make such a statement it needs considerably more explanation and analysis, as there seems no basis to estimate the changes in sources well enough to make such a statement. vi) It seems that GCMs agree with a small change in accumulation rate.

Taking all these estimates together with the one I derived from Kapsner et al, and noting that they all have significant weaknesses, I see no reason to rule out a very small change in accumulation rate, which is suggested by Kapsner et al, approach vi, and the lower range of the Buchardt approach when you include their NE data. It's unfortunate but it means you simply can't constrain accumulation rate this way.

Page 14, line 28. I don't see the relevance of this: at current mean summer temperatures we know we can get melt layers (as we did this year). You certainly don't need the summer mean to be 5 degrees warmer than present to expect significant melt every

year.

Fig 1. Please remove the age scale for the shaded portion beyond 128 ka, as we have no basis for it. Summer insolation should also be cut at 128 ka as it cannot be compared to the climate curves below it beyond that age.

Figure 3. Please make this figure clearer in the caption. I suggest something like: "The black circles….NEEM. The light curves are contours of accumulation rate-temperature combinations for a given value if del15N, with that value shown in the curve. The coloured examples correspond to the measured LIG del15N for GRIP (blue), NGRIP (purple) and NEEM (red). The horizontal lines and darker curves correspond to different estimates of the accumulation rate (for the curves, this is as a function of temperature)." Then having explained the overall point of the diagram, you can explain each curve, but of course I would hope that the Buchardt lines will be shown and will include a wider range than in the text, that the Kapsner line is added, and that the vertical arrows better reflect the range of credible estimates.
* * *

---

## Author Comment (AC1) · 7 Jun 2016

The comments of the two reviewers and M. Löfverström are very helpful and we thank them for this careful reading. They clearly show parts of the manuscript that need to be improved and rewritten.

The main point is probably the main conclusion of the paper and the warm LIG temperature at NEEM. First, we agree that our results are probably not presented the right way and that our conclusion should be rewritten taking into account in figure 3 the broad ranges of accumulation estimate from modeling and data. This will lower the lowest LIG temperature for the NEEM deposition site by 1°C as noted by E. Wolff. Second, another important point to take into account for this conclusion is the surprising comparison between NGRIP and NEEM d15N level that leads to different estimates of the

LIG temperature. It should be noted that the value of 0.29 permil taken for NGRIP for the LIG in the table 1 was not representative enough of the NGRIP LIG section and we apologize for this mistake. In the previous manuscript, we have taken the deeper individual d15N value from the NGRIP ice core record from the initial low resolution study of Landais et al. (2005) while we should have taken the average of d15N values from the high resolution study of Capron et al. (GRL, 2012) over the bottom part of the NGRIP ice core corresponding to the oldest 1000 year period in the NGRIP ice core. When correcting for this and making an average over 1000 years, we obtain a mean d15N value of 0.275 permil for NGRIP LIG. This corresponds to a LIG mean temperature of -26.5+/-2°C, hence an increase of temperature of 5+/-2°C compared to pre-industrial temperature. Even if this temperature increase is larger than in the previous manuscript, this is still smaller than the temperature increase estimated at the NEEM deposition site. As suggested by reviewer 1, we will thus add a discussion about this difference. This discussion will include the following arguments: - Difficulty to precisely date the bottom of the NGRIP ice core because of the lack of precise relative and absolute age markers. - No evidence of melt-layers at NGRIP compared to NEEM for the LIG part, - At NEEM, surface deposition site and current drilling site have significant different surface conditions today. This is not the case for NGRIP.

Finally, we will also provide some more details on the modeling outputs and how they can be used to document seasonality of precipitation and possible link with d18O.

We provide an answer to every individual comment below:

"Comments by M. Löfverström

1: I am somewhat critical to how you use climate model data in the current version of the paper. First, all models are to varying degree simplified versions of reality so a perfect match with proxy data is not to be expected. Secondly, comparing proxies with data in a single grid cell is arguably a misuse of climate models as they are designed to give an indication of the average conditions over a large region (the size of the region

is dependent on the model complexity and grid resolution but a rule of thumb is to use at least a few grid cells). You should perhaps also comment on the range of models used in Fig. 4; the figure presents data from EMICS to full GCMs, which are worlds apart both in terms of complexity and modeling strategy (e.g. data constrained vs. free running, highly simplified vs. very complex, etc)."

» Indeed the models we used have very different complexities and resolution. We mention that on page 12, line 17 and will emphasize it further in the next version. This will help to clarify the model – data comparison which is indeed not easy for the NEEM site as mentioned in the first manuscript. As for the grid resolution, we would argue to keep the climate model-data comparison for just the NEEM upstream grid box. This grid box is 3.75x2.5 degrees, "regridded" sometimes from lower resolution climate models. For answering this comment, we did the same analysis for the surrounding grid boxes, and the mean over the 9 grid boxes (see pdf attached). The results change indeed a bit, but not drastically, and the assumed maximum of $2°C$ from LIG external forcing is still valid. Another argument for keeping this grid bow is that the temperature patterns over Greenland are relatively smooth, so you would also not expect the results to change much when analyzing another grid box. This can already be seen in the temperature maps of the Bakker and Lunt papers quoted in the manuscript.

"2: You mention that the _18O signal recorded in ice cores can be influenced by changes in transport pathways and precipitation seasonality. The former is a bit tricky to investigate but you can easily perform a similar analysis as in Pausata and Löfverström (2015) (On the enigmatic similarity in Greenland _18O between the Oldest and Younger Dryas, Geophys. Res. Letters, 42, doi:10.1002/2015GL066042) and quantify the importance of precipitation seasonality and cloud temperature for the implied _18O signal in the models."

» This comment is not easy to answer within the scope of this study because we do not have monthly mean model results for all models. Moreover, it would make more sense for this question to use models equipped with water isotopes.

Still, we had a look on the NorESM simulations using available surface temperature and precipitation rate at a monthly resolution (unfortunately, cloud temperatures are not available). If we calculate the temperature weighted by the precipitation rate for both pre-industrial and LIG, we find a much more important increase (roughly a factor of two) between pre-industrial and LIG than when using the mean annual temperature. Indeed, both the temperature and accumulation rate seasonalities are different from pre-industrial during LIG at NEEM: summer temperature (accumulation rate) increases by 3.5°C (7 mm/month) and winter temperature (accumulation rate) decreases by 2°C (3 mm/month). This monthly temperature and accumulation rate patterns are however significantly different from one grid box to another and probably from one model to another. We thus propose to include this discussion in a revised manuscript in a short paragraph indicating that this result is strongly variable and that a proper study of the seasonality effect on d18O should be made with a model equipped with water isotopes.

"3: I would be careful citing unpublished work or papers in open discussion, except of course if the papers are accepted and about to be released. There is never a guarantee that a paper will be accepted only because it is in review and the methodology and conclusions might change significantly when the paper is finally published."

» This will be done in a revised manuscript. One paper which is now in major revision will be removed from the reference list.

- Comments by Anonymous Referee #1

"Major points: p11 It is very interesting that the NEEM and NGRIP d15N reconstructions at 120 ka are so different: +8.5 versus +3.2C. This merits more analysis and attention. In particular, is the 120 ka NGRIP is less affected by uncertainty generated by melt processes? This should be mentioned in the conclusions and in the abstract. Moreover the likely reasons for this discrepancy should be subject to further investigation/discussion and emphasis within this manuscript."

» This is indeed important to focus more on the NGRIP ice core which was kind of

neglected in the first version. When coming back to these data, we actually realized that the d15N value of 0.29 permil taken for NGRIP for the LIG in the table 1 was not representative enough of the NGRIP LIG section and we apologize for this mistake. In the previous manuscript, we have taken the deeper individual d15N value from the NGRIP ice core record from the initial low resolution study of Landais et al. (2005) while we should have taken the average of d15N values from the high resolution study of Capron et al. (GRL, 2012) over the bottom part of the NGRIP ice core corresponding to the oldest 1000 year period in the NGRIP ice core. When correcting for this and making an average over 1000 years, we obtain a mean d15N value of 0.275 permil for NGRIP LIG. This corresponds to a LIG mean temperature of -26.5+-2°C, hence an increase of temperature of 5+-2°C compared to pre-industrial temperature. Even if this temperature increase is larger than in the previous manuscript, this is still less than the temperature increase estimated at the NEEM deposition site. As a consequence and as suggested by reviewer 1, we will add a discussion about this difference. This discussion will include the following arguments: - Difficulty to precisely date the bottom of the NGRIP ice core because of the lack of precise dating constraints. In the new text, we will particularly mention the difficulty to produce precise gas synchronization with other dated records (in particular due to the high CH4 values) and also the lack of other independent precise absolute age markers (see also Veres et al. CP 2013 and Govin et al. QSR 2015) - No evidence of melt-layers at NGRIP compared to NEEM for the LIG part - At NEEM, surface deposition site and current drilling site have significant different surface conditions today. This is not the case for NGRIP.

"p12 L7-9 Have the studies by Metz et al not already provided enough information to attempt to constrain these 'other influences' on surface temperature and accumulation rates? And if so, should this not be included within the analysis/uncertainties for this study?"

» This is a good point, in line with the comments of E. Wolff below and we will include the uncertainty on accumulation in the new determination which will lower the low range

of LIG temperature increase at the NEEM deposition site to 6°C (instead of 7°C).

Minor points:

» All minor points will be taken into account in the revised manuscript.

Comments by EW Wolff (Referee)

"the authors reach a conclusion that I don't think their data justify, especially if one includes a further estimate of the accumulation rate, which I would rate as being at least as valid (perhaps more so) compared to the ones they choose. The critical part of the paper is Figure 3. I would make a first comment that, given how crucial this figure is, it's extremely hard to follow. I'll follow this up later. If I accept all the accumulation estimates shown (but I don't, see below), I reach the conclusion of the authors in section 3.2.3, but not the overall conclusion of the paper. This section seems to conclude: For Summit, the data can probably not be interpreted as a pure thermal signal; For NEEM, you seem to choose a range between the +20% accumulation (4.5 degrees warming), and the "M-D" approach (_8 degrees warming). The authors don't explicitly state how they corrected for the upstream issue but I assume from Table 2 that they added 2.5 degrees. Thus the range they estimate is 7-10.5 degrees. For NGRIP, the best estimate is 3.2+/-0.7 degrees, with no significant upstream correction, and a possible extra 0.5 degrees for the warmest part of the LIG. So taken at face value, one would conclude that one site gives 7-10.5 degrees and the other gives 2.5-4 degrees, which does not seem like a basis for asserting that the climate was 8 degrees warmer and that the NEEM paradox is confirmed."

» We agree that the conclusion should be revised and we certainly did not focus enough on the NGRIP site in the previous manuscript (and too much on the NEEM site). Also, as previously mentioned, we have realized only after we submitted our manuscript that the LIG d15N value taken for NGRIP was not correct. When accounting for the updated LIG d15N value based on Capron et al. 2012 high resolution d15N record, we obtain a LIG temperature increase at the NGRIP deposition site larger than originally calculated. In addition, the arguments developed below are sound as well and we should not discard lowest accumulation rate increases as performed in the manuscript, especially for NEEM whose deposition site is not very well known. This will be corrected in the new version. The temperature of the LIG will thus be higher by 3-7°C at NGRIP and 6 to 11°C at the NEEM deposition site relative to preindustrial (11°C being highly improbable as already explained in the first manuscript and in agreement with comments below). It is still a bit high compared to the model simulations but the conclusion will be tuned down compared to the first version.

"However, what Figure 3 really tells us is that there is no good way of estimating the accumulation rate in the LIG, and therefore in the end we can't constrain the temperature this way. I can suggest another very valid way to estimate the accumulation rate. In Kapsner et al (1995), which the authors cite, an accumulation rate-temperature estimate for the early Holocene (pre-Boreal, which seems the most relevant) is made from correlation of annual layer thicknesses with oxygen isotope ratio measurements. They find a rather shallow slope within climate periods such as the pre-Boreal. Because they actually did a regression of accumulation against oxygen isotopes, we can in fact estimate directly (without going through temperature) what should be the acc rate change for an oxygen isotope change of 3.5 permil (Table 2, change at deposition site). Given that they report a slope of 0.9%/K, and they used a calibration of 0.53 permil/K, their acc-isotope slope must have been 1.7%/permil, and a 3.6 permil change would correspond to a 6% change in accumulation rate. This would suggest a 3.5 degree warmer firn column, which adding the 2.5 degrees upstream correction, implies the NEEM region was 6 degrees warmer than present. "

» This is indeed correct. It is still surprising that the results from Kapsner et al. (1995) for GISP2 are different from the Dahl-Jensen et al. (1993) for GRIP and Buchardt et al. (2012) while the methods are similar. But this goes in line with the other comments that the accumulation rate vs temperature relationship may be very variable in Greenland and we should not have discarded the possible stable accumulation rate scenario for

warmer period. This will be corrected. We also understand the need for simplification of figure 3 where we can indeed only show the LIG accumulation rate possible range and remove the accumulation rate vs temperature relationships. Figure 3 will be redrawn together with the text on accumulation rate estimates.

"In the rest of this review I will go through the paper in more detail, and comment some more on Figure 3. However my overall suggestion is that, taking into account another realistic way of calculating accumulation rate, and the NGRIP result, the conclusions and abstract should be much more balanced and should not claim to be confirming the very high estimate of NEEM Project Members."

» As mentioned above, this will indeed be done both in the abstract and main text. For the following detailed comments, we answer here only to the major comments and all minor comments will be taken into account.

"As I already suggested, you really only show in this section that we have no good basis for estimating the accumulation rate that applies to this warm interglacial. Here I just comment on some of the estimates you give: i) The Kapsner paper quite conclusively shows that the thermodynamic approach is not really applicable to a situation like that in Greenland, where most of the precipitation is related to cyclones and storm activity. It's fine to mention it as an option, but it is clearly flawed. When it comes to figure 3, I really don't understand what the grey shaded area is meant to represent, so if you leave it, it needs a better explanation. ii) It seems as if this Benson approach is actually an empirical spatial version of the thermodynamic approach? However, while I could almost justify this approach for single storms tracked (and their integrated effect over a year) spatially across Greenland, there seems no basis for translating that into the temporal domain, as you acknowledge on Page 8, line 22. Note also that Benson 1962 seems missing from the reference list."

» For simplicity, we will remove the Benson 1962 outputs (indeed a temporal relationship) from Figure 3. The idea is still to keep it as a reference because it is still used

today for validation of accumulation rate estimate in Greenland.

" iii) Same applies for the Buchhardt approach, but since the blue lines are missing from the plot I cannot assess it. I understand why you prefer the sensitivity Buchardt derived for the NW region but don't understand why you give a range for Greenland of 6.7-9.6%/K, when Buchardt's table gives a value of only 1.5%/K (with an error bar encompassing zero) for the nearby NE region, and even a negative slope for SW Greenland. I do not think you have summarised the Buchhardt outcome fairly; in my opinion it gives weak evidence for a 6.7%/K slope, but with an uncertainty that encompasses a zero slope. Note also that it is unnecessary to draw a curve to represent Buchardt; he actually derived an acc rate-del18O slope, so for a measured change in del18O, you can pinpoint precisely the range of the change in acc rate and draw them as horizontal lines . "

» We will include the NW region as well in the discussion and simplify figure 3 as mentioned above (only give the range of possible accumulation rate as for 10Be or chemistry)

"iv) I have not checked the MD approach specifically, but note that this is the same paper that derived a much larger del18O-temperature slope and therefore a smaller temperature change than NEEM Project Members, so it seems inconsistent to use the same model simulations to derive the opposite result."

» The accumulation – temperature relationships given in Masson-Delmotte et al. (2015) have indeed a very high slope for the recent warming. This is not necessary inconsistent with the high d18O vs temperature slope but we understand that it may lead to unnecessary complications. We thus propose to gather the accumulation vs temperature and d18O vs temperature relationship from same source (model outputs, reanalyses, . . .) and only give the final accumulation rate vs d18O relationship (hence combining values from Tables 6 and 7 from Masson-Delmotte et al., 2015). This leads to values around 10 %.‰1, hence a 35% change in accumulation rate for an increase in d18O

of 3.5‰ at the NEEM deposition site. This again leads to a temperature reconstruction for the NEEM deposition site at the upper boundary of the 6-11°C range.

" v) I have not checked the 10Be estimates, but note only that because deposition in Greenland under interglacial conditions is completely dominated by wet deposition, it seems very unlikely that one can derive accumulation rates from 10Be. I agree with your implication that the change in overall scavenging would have to be accounted for in a sophisticated model, and I don't believe that local accumulation rate would be the dominant control on 10Be concentration or flux. For the same reason, I do not understand at all your statement on Page 9, lines 22-24 that changes in aerosol rule out no change in accumulation: if you want to make such a statement it needs considerably more explanation and analysis, as there seems no basis to estimate the changes in sources well enough to make such a statement. "

» The argument for the statement that the aerosol chemistry renders the same accumulation in the Eemian and the Holocene as unlikely is as follows: If we compare aerosol species which are dominated by wet deposition such as Na (sea salt aerosol) and NO3 (lightning activity, biological activity) we see that the concentration in the ice in the Eemian is drastically lower than in the HOL (only about 50%). As these species are mainly wet deposited this is only possible if the atmospheric aerosol concentrations over the ice sheet was also lower by 50% at that time. It is unlikely that both Na and NO3 (which have completely different sources and transport pathways) have both a 50% reduction of source emissions, which is too large anyway. We can also get a reduction of 50% in atmospheric Na and NO3 concentrations over the ice, if we increase the precipitation rate along the transport pathway and thus increase wet deposition en route. So what the chemistry points evidences is that the precipitation rate during transport in the Northern Hemisphere was significantly higher in the Eemian than in the HOL. If this is not case on the Greenland ice sheet, it means that the precipitation rate should be higher in the Eemian everywhere else except in Greenland, which is unlikely. In particular as sea salt aerosol transport from the open ocean comes jointly

with water vapor transport to the ice sheet through storm events an increased washout of Na en route is expected due to higher precipitation rates and this would also dump more water on the ice sheet. Finally, note that if we scale the change in precipitation rate en route and in local accumulation rate from the HOL to the Eemian according to the Buchardt formula, no change in Na and NO3 sources strength is required at all. Still, we will include the reconstruction with no change in accumulation as suggested by E. Wolff. It indeed makes sense.

"vi) It seems that GCMs agree with a small change in accumulation rate. Taking all these estimates together with the one I derived from Kapsner et al, and noting that they all have significant weaknesses, I see no reason to rule out a very small change in accumulation rate, which is suggested by Kapsner et al, approach vi, and the lower range of the Buchardt approach when you include their NE data. It's unfortunate but it means you simply can't constrain accumulation rate this way."

» This is correct. It will be included as stated above

"Page 14, line 28. I don't see the relevance of this: at current mean summer temperatures we know we can get melt layers (as we did this year). You certainly don't need the summer mean to be 5 degrees warmer than present to expect significant melt every year."

» We were referring to comparison with pre-industrial temperature. In the recent years, melt-layers were observed at NEEM during heat wave as in 2012 with summer temperature of $\sim5°C$ warmer than pre-industrial summers. We will rewrite this sentence to clarify our statement.

Please also note the supplement to this comment:
http://www.clim-past-discuss.net/cp-2016-28/cp-2016-28-AC1-supplement.pdf

**Supplement:**

**Annual mean temp anom around NEEMup (77.5:72.5N,48.75:41.25W)**

[Figure]

[Figure]

**Annual mean temp anom around NEEMup (77.5N,48.75W)**

Legend:
- Bern3D
- CCSM3
- CLIMBER2
- COSMOS
- CSIRO
- FAMOUS
- KCM
- LOVECLIM
- MPI-UW
- HADCM3_Bris
- CCSM3_Bremen
- CCSM3_LNN
- CCSM3_NCAR
- COSMOS_AWI
- KCM_Kiel
- LOVECLIM_Ams
- LOVECLIM_LLN
- MIROC_Tokyo
- CLIMBER_LSCE
- IPSL_LSCE
- CSIRO_UNSW
- NORESM_BCCR

[Figure]

**Annual mean temp anom around NEEMup (77.5N,45.00W)**

Legend:
- Bern3D
- CCSM3
- CLIMBER2
- COSMOS
- CSIRO
- FAMOUS
- KCM
- LOVECLIM
- MPI-UW
- HADCM3_Bris
- CCSM3_Bremen
- CCSM3_LNN
- CCSM3_NCAR
- COSMOS_AWI
- KCM_Kiel
- LOVECLIM_Ams
- LOVECLIM_LLN
- MIROC_Tokyo
- CLIMBER_LSCE
- IPSL_LSCE
- CSIRO_UNSW
- NORESM_BCCR

Axes: Annual mean temp anom to PI [degC] (−5 to 5); Time [ka] (115 to 130)

[Figure]

**Annual mean temp anom around NEEMup (77.5N,41.25W)**

Legend:
- Bern3D
- CCSM3
- CLIMBER2
- COSMOS
- CSIRO
- FAMOUS
- KCM
- LOVECLIM
- MPI-UW
- HADCM3_Bris
- CCSM3_Bremen
- CCSM3_LNN
- CCSM3_NCAR
- COSMOS_AWI
- KCM_Kiel
- LOVECLIM_Ams
- LOVECLIM_LLN
- MIROC_Tokyo
- CLIMBER_LSCE
- IPSL_LSCE
- CSIRO_UNSW
- NORESM_BCCR

Axis labels: Annual mean temp anom to PI [degC]; Time [ka]

[Figure]

**Annual mean temp anom around NEEMup (75.0N,48.75W)**

Legend:
- Bern3D
- CCSM3
- CLIMBER2
- COSMOS
- CSIRO
- FAMOUS
- KCM
- LOVECLIM
- MPI-UW
- HADCM3_Bris
- CCSM3_Bremen
- CCSM3_LNN
- CCSM3_NCAR
- COSMOS_AWI
- KCM_Kiel
- LOVECLIM_Ams
- LOVECLIM_LLN
- MIROC_Tokyo
- CLIMBER_LSCE
- IPSL_LSCE
- CSIRO_UNSW
- NORESM_BCCR

Axis labels: Annual mean temp anom to PI [degC], Time [ka]

[Figure]

**Annual mean temp anom around NEEMup (75.0N,45.00W)**

Legend:
- Bern3D
- CCSM3
- CLIMBER2
- COSMOS
- CSIRO
- FAMOUS
- KCM
- LOVECLIM
- MPI-UW
- HADCM3_Bris
- CCSM3_Bremen
- CCSM3_LNN
- CCSM3_NCAR
- COSMOS_AWI
- KCM_Kiel
- LOVECLIM_Ams
- LOVECLIM_LLN
- MIROC_Tokyo
- CLIMBER_LSCE
- IPSL_LSCE
- CSIRO_UNSW
- NORESM_BCCR

Axis labels: Time [ka] (x-axis), Annual mean temp anom to PI [degC] (y-axis)

[Figure]

**Annual mean temp anom around NEEMup (75.0N,41.25W)**

Legend:
- Bern3D
- CCSM3
- CLIMBER2
- COSMOS
- CSIRO
- FAMOUS
- KCM
- LOVECLIM
- MPI-UW
- HADCM3_Bris
- CCSM3_Bremen
- CCSM3_LNN
- CCSM3_NCAR
- COSMOS_AWI
- KCM_Kiel
- LOVECLIM_Ams
- LOVECLIM_LLN
- MIROC_Tokyo
- CLIMBER_LSCE
- IPSL_LSCE
- CSIRO_UNSW
- NORESM_BCCR

X-axis: Time [ka]
Y-axis: Annual mean temp anom to PI [degC]

[Figure]

**Annual mean temp anom around NEEMup (72.5N,48.75W)**

Legend:
- Bern3D
- CCSM3
- CLIMBER2
- COSMOS
- CSIRO
- FAMOUS
- KCM
- LOVECLIM
- MPI-UW
- HADCM3_Bris
- CCSM3_Bremen
- CCSM3_LNN
- CCSM3_NCAR
- COSMOS_AWI
- KCM_Kiel
- LOVECLIM_Ams
- LOVECLIM_LLN
- MIROC_Tokyo
- CLIMBER_LSCE
- IPSL_LSCE
- CSIRO_UNSW
- NORESM_BCCR

Axis labels: Annual mean temp anom to PI [degC], Time [ka]

[Figure]

**Annual mean temp anom around NEEMup (72.5N,45.00W)**

Legend:
- Bern3D
- CCSM3
- CLIMBER2
- COSMOS
- CSIRO
- FAMOUS
- KCM
- LOVECLIM
- MPI-UW
- HADCM3_Bris
- CCSM3_Bremen
- CCSM3_LNN
- CCSM3_NCAR
- COSMOS_AWI
- KCM_Kiel
- LOVECLIM_Ams
- LOVECLIM_LLN
- MIROC_Tokyo
- CLIMBER_LSCE
- IPSL_LSCE
- CSIRO_UNSW
- NORESM_BCCR

x-axis: Time [ka]
y-axis: Annual mean temp anom to PI [degC]

[Figure]

**Annual mean temp anom around NEEMup (72.5N,41.25W)**

Legend:
- Bern3D
- CCSM3
- CLIMBER2
- COSMOS
- CSIRO
- FAMOUS
- KCM
- LOVECLIM
- MPI-UW
- HADCM3_Bris
- CCSM3_Bremen
- CCSM3_LNN
- CCSM3_NCAR
- COSMOS_AWI
- KCM_Kiel
- LOVECLIM_Ams
- LOVECLIM_LLN
- MIROC_Tokyo
- CLIMBER_LSCE
- IPSL_LSCE
- CSIRO_UNSW
- NORESM_BCCR

Y-axis: Annual mean temp anom to PI [degC]
X-axis: Time [ka]

---

## Author Response (AR1)

The comments of the two reviewers and M. Löfverström are very helpful and we thank them for this careful reading. They clearly show parts of the manuscript that need to be improved and rewritten.

The main point is probably the main conclusion of the paper and the warm LIG temperature at NEEM.
First, we agree that our results are probably not presented the right way and that our conclusion should be rewritten taking into account in figure 3 the broad ranges of accumulation estimate from modeling and data. This will lower the lowest LIG temperature for the NEEM deposition site by 1°C as noted by E. Wolff.
Second, another important point to take into account for this conclusion is the surprising comparison between NGRIP and NEEM d15N level that leads to different estimates of the LIG temperature. It should be noted that the value of 0.29 permil taken for NGRIP for the LIG in the table 1 was not representative enough of the NGRIP LIG section and we apologize for this mistake. In the previous manuscript, we have taken the deeper individual d15N value from the NGRIP ice core record from the initial low resolution study of Landais et al. (2005) while we should have taken the average of d15N values from the high resolution study of Capron et al. (GRL, 2012) over the bottom part of the NGRIP ice core corresponding to the oldest 1000 year period in the NGRIP ice core. When correcting for this and making an average over 1000 years, we obtain a mean d15N value of 0.275 permil for NGRIP LIG. This corresponds to a LIG mean temperature of -26.5+/-2°C, hence an increase of temperature of 5+/-2°C compared to pre-industrial temperature. Even if this temperature increase is larger than in the previous manuscript, this is still smaller than the temperature increase estimated at the NEEM deposition site. This has been corrected in the figure 3. Then, as suggested by reviewer 1, we have added a discussion about this difference, especially on p. 12 and 13.

Finally, we have also provided some more details on the modeling outputs and how they can be used to document seasonality of precipitation and possible link with d18O. The corresponding additions are in yellow highlights in section 3.3.

We provide an answer to every individual comment below:

**Comments by M. Löfverström**

1: I am somewhat critical to how you use climate model data in the current version of the paper. First, all models are to varying degree simplified versions of reality so a perfect match with proxy data is not to be expected. Secondly, comparing proxies with data in a single grid cell is arguably a misuse of climate models as they are designed to give an indication of the average conditions over a large region (the size of the region is dependent on the model complexity and grid resolution but a rule of thumb is to use at least a few grid cells). You should perhaps also comment on the range of models used in Fig. 4; the figure presents data from EMICS to full GCMs, which are worlds apart both in terms of complexity and modeling strategy (e.g. data constrained vs. free running, highly simplified vs. very complex, etc).

Indeed the models we used have very different complexities and resolution. We mention that on page 12, line 17 in the previous manuscript and emphasize it further in the next version (section 3.3). We hope that this helps to clarify the model – data comparison.

As for the grid resolution, we would argue to keep the climate model-data comparison for just the NEEM upstream grid box. This grid box is 3.75x2.5 degrees, "regridded" sometimes from lower resolution climate models. For answering this comment, we did the same analysis for the surrounding grid boxes, and the mean over the 9 grid boxes (see answer to comments in the online discussion version). The results change indeed a bit, but not drastically, and the assumed maximum of 2°C from LIG external forcing is still valid.

Another argument for keeping this grid bow is that the temperature patterns over Greenland are relatively smooth, so you would also not expect the results to change much when analyzing another grid box. This can already be seen in the temperature maps of the Bakker and Lunt papers quoted in the manuscript.

2: You mention that the _18O signal recorded in ice cores can be influenced by changes in transport pathways and precipitation seasonality. The former is a bit tricky to investigate but you can easily perform a similar analysis as in Pausata and Löfverström (2015) (On the enigmatic similarity in Greenland _18O between the Oldest and Younger Dryas, Geophys. Res. Letters, 42, doi:10.1002/2015GL066042) and quantify the importance of precipitation seasonality and cloud temperature for the implied _18O signal in the models.

This comment is not easy to answer within the scope of this study because we do not have monthly mean model results for all models. Moreover, it would make more sense for this question to use models equipped with water isotopes.

Still, we had a look on the NorESM simulations using available surface temperature and precipitation rate at a monthly resolution (unfortunately, cloud temperatures are not available). If we calculate the temperature weighted by the precipitation rate for both pre-industrial and LIG, we find a much more important increase (roughly a factor of two) between pre-industrial and LIG than when using the mean annual temperature. Indeed, both the temperature and accumulation rate seasonalities are different from pre-industrial during LIG at NEEM: summer temperature (accumulation rate) increases by 3.5°C (7 mm/month) and winter temperature (accumulation rate) decreases by 2°C (3 mm/month). This monthly temperature and accumulation rate patterns are however significantly different from one grid box to another and probably from one model to another. We thus propose to include this discussion in a revised manuscript in a short paragraph indicating that this result is strongly variable and that a proper study of the seasonality effect on d18O should be made with a model equipped with water isotopes. The corresponding text is insert on p.4:

"As an example, if we use surface temperature and precipitation rate in monthly resolution from the NorESM model at the NEEM LIG deposition site, we observe a simulated increase in summer temperature (accumulation rate) by 3.5°C (7 mm/month) and a decreased in winter temperature (accumulation rate) by 2°C (3 mm/month).

3: I would be careful citing unpublished work or papers in open discussion, except of course if the papers are accepted and about to be released. There is never a guarantee that a paper will be accepted only because it is in review and the methodology and conclusions might change significantly when the paper is finally published.

We have removed the reference to Langebroek and Nisancioglu, TCD

- **Comments by Anonymous Referee #1**

Major points:
p11 It is very interesting that the NEEM and NGRIP d15N reconstructions at 120 ka are so different: +8.5 versus +3.2C. This merits more analysis and attention. In particular, is the 120 ka NGRIP is less affected by uncertainty generated by melt processes?
This should be mentioned in the conclusions and in the abstract. Moreover the likely reasons for this discrepancy should be subject to further investigation/discussion and emphasis within this manuscript.

This is indeed important to focus more on the NGRIP ice core which was neglected in the first version. When coming back to these data, we actually realized that the d15N value of 0.29 permil taken for NGRIP for the LIG in the table 1 was not representative enough of the NGRIP LIG section and we apologize for this mistake. In the previous manuscript, we have taken the deeper individual d15N value from the NGRIP ice core record from the initial low resolution study of Landais et al. (2005) while we should have taken the average of d15N values from the high resolution study of Capron et al. (GRL, 2012) over the bottom part of the NGRIP ice core corresponding to the oldest 1000 year period in the NGRIP ice core. When correcting for this and making an average over 1000 years, we obtain a mean d15N value of 0.275 permil for NGRIP LIG. This corresponds to a LIG mean temperature of -26.5+-2°C, hence an increase of temperature of 5+-2°C compared to pre-industrial temperature. This has been corrected all along the text and in Figure 3. Even if this temperature increase is larger than in the previous manuscript, this is still less than the temperature increase estimated at the NEEM deposition site. As a consequence and as suggested by reviewer 1, a discussion is added on p. 12-13:

"In summary, the NGRIP LIG vs pre-industrial temperature increase (+5.2±2.3°C) is thus on the lower end but still compatible within error bars with the NEEM LIG vs pre-industrial temperature increase (+8±2.5°C at 120 ka using the aforementioned $\delta^{15}N$ value of 0.256 ‰). Several explanations can explain this discrepancy. First, the dating of the NGRIP and NEEM bottom parts is difficult because of the lack of precise relative and absolute age markers; this limits our confidence that we indeed compare $\delta^{15}N$ levels of same age on the two different cores. Second, we do not have any evidence of melt layers in the bottom of the NGRIP core, opposite to NEEM. This suggests that the NEEM deposition site was indeed warmer than NGRIP at LIG but also suggests that firn densification may have been affected by this process at NEEM which

would bias our reconstruction. Finally, our temperature reconstruction at NEEM is complicated by the fact that the NEEM LIG deposition site and NEEM drilling site have different surface conditions. In particular, an estimate of the pre-industrial accumulation rate is missing at the NEEM deposition site."

p12 L7-9 Have the studies by Metz et al not already provided enough information to attempt to constrain these 'other influences' on surface temperature and accumulation rates? And if so, should this not be included within the analysis/uncertainties for this study?

This is a good point, in line with the comments of E. Wolff below and we have included the uncertainty on accumulation in the new determination which lowers the low range of LIG temperature increase at the NEEM deposition site to 6°C (instead of 7°C). Section 3.2.1 has thus been almost entirely rewritten (see text with yellow highlights at the end of this document).

Minor points:

All minor points have been taken into account in the revised manuscript.

**Comments by EW Wolff (Referee)**

the authors reach a conclusion that I don't think their data justify, especially if one includes a further estimate of the accumulation rate, which I would rate as being at least as valid (perhaps more so) compared to the ones they choose. The critical part of the paper is Figure 3. I would make a first comment that, given how crucial this figure is, it's extremely hard to follow. I'll follow this up later. If I accept all the accumulation estimates shown (but I don't, see below), I reach the conclusion of the authors in section 3.2.3, but not the overall conclusion of the paper. This section seems to conclude:
For Summit, the data can probably not be interpreted as a pure thermal signal; For NEEM, you seem to choose a range between the +20% accumulation (4.5 degrees warming), and the "M-D" approach (_8 degrees warming). The authors don't explicitly state how they corrected for the upstream issue but I assume from Table 2 that they added 2.5 degrees. Thus the range they estimate is 7-10.5 degrees. For NGRIP, the best estimate is 3.2+/-0.7 degrees, with no significant upstream correction, and a possible extra 0.5 degrees for the warmest part of the LIG.
So taken at face value, one would conclude that one site gives 7-10.5 degrees and the other gives 2.5-4 degrees, which does not seem like a basis for asserting that the climate was 8 degrees warmer and that the NEEM paradox is confirmed.

We agree that the conclusion should be revised and we certainly did not focus enough on the NGRIP site in the previous manuscript (and too much on the NEEM site). Also, as previously mentioned, we have realized only after we submitted our manuscript that the LIG d15N value taken for NGRIP was not correct. When accounting for the updated LIG d15N value based on Capron et al. 2012 high resolution d15N record, we obtain a LIG temperature increase at the NGRIP deposition site larger than originally calculated. In addition, the arguments developed below are sound as well and we should not discard lowest accumulation rate increases as

performed in the manuscript, especially for NEEM whose deposition site is not very well known. This will be corrected in the new version. The temperature of the LIG will thus be higher by 3-7°C at NGRIP and 6 to 11°C at the NEEM deposition site relative to preindustrial (11°C being highly improbable as already explained in the first manuscript and in agreement with comments below). It is still a bit high compared to the model simulations but the conclusion has been tuned down compared to the first version (see section 4 in the text at the end of this document with many parts removed and parts rewritten in yellow highlights"

However, what Figure 3 really tells us is that there is no good way of estimating the accumulation rate in the LIG, and therefore in the end we can't constrain the temperature this way. I can suggest another very valid way to estimate the accumulation rate.
In Kapsner et al (1995), which the authors cite, an accumulation rate-temperature estimate for the early Holocene (pre-Boreal, which seems the most relevant) is made from correlation of annual layer thicknesses with oxygen isotope ratio measurements. They find a rather shallow slope within climate periods such as the pre-Boreal. Because they actually did a regression of accumulation against oxygen isotopes, we can in fact estimate directly (without going through temperature) what should be the acc rate change for an oxygen isotope change of 3.5 permil (Table 2, change at deposition site). Given that they report a slope of 0.9%/K, and they used a calibration of 0.53 permil/K, their acc-isotope slope must have been 1.7%/permil, and a 3.6 permil change would correspond to a 6% change in accumulation rate. This would suggest a 3.5 degree warmer firn column, which adding the 2.5 degrees upstream correction, implies the NEEM region was 6 degrees warmer than present.

This is indeed correct. It is still surprising that the results from Kapsner et al. (1995) for GISP2 are different from the Dahl-Jensen et al. (1993) for GRIP and Buchardt et al. (2012) while the methods are similar. But this goes in line with the other comments that the accumulation rate vs temperature relationship may be very variable in Greenland and we should not have discarded the possible stable accumulation rate scenario for warmer period. This has been corrected. We also understand the need for simplification of figure 3 where we can indeed only show the LIG accumulation rate possible range and remove the accumulation rate vs temperature relationships. Figure 3 has be redrawn together with the text on accumulation rate estimates.

Part of section 3.2.1 rewritten for the Kapsner et al., 1995 reference:
"iii: Based on the GISP2 ice core records over the last deglaciation, Kapsner et al. (1995) showed that the relationship between Greenland accumulation rate and temperature was not stable because of variations in atmospheric circulation. Still, they were able to propose a temporal relationship between accumulation rate based on annual layer counting and $\delta^{18}O$, or temperature reconstructed from $\delta^{18}O$ and a calibration based on borehole temperature measurements (leading to a $\delta^{18}O$ vs temperature slope of 0.53‰.°C$^{-1}$). The inferred sensitivity of snow accumulation rate to temperature change during interglacial period varies from 0.9%.°C$^{-1}$ (Holocene) to 7.5%.°C$^{-1}$ (Bølling-Allerød), with an uncertainty encompassing zero.
More recently, Buchardt et al. (2012) followed a similar approach but using numerous ice cores. They used an array of 52 shallow ice cores spanning the last decades to centuries with accumulation rate estimates from annual layer

counting on $\delta^{18}O_{ice}$ profiles. They identified different temperature vs accumulation rate relationships from one region to another. In central and north Greenland corresponding to the location of the NEEM, NGRIP, GRIP and GISP2 deep ice cores, the Buchardt approach suggests a sensitivity of 1.5 to 9.4%.°C$^{-1}$ with an uncertainty encompassing zero. This sensitivity is obtained with a $\delta^{18}O$ vs temperature sensitivity slope of 0.67‰.°C$^{-1}$ so that the Kapsner and Buchardt estimates agree on a 0 to 14%.‰$^{-1}$ accumulation rate vs $\delta^{18}O$ sensitivity. "

In the rest of this review I will go through the paper in more detail, and comment some more on Figure 3. However my overall suggestion is that, taking into account another realistic way of calculating accumulation rate, and the NGRIP result, the conclusions and abstract should be much more balanced and should not claim to be confirming the very high estimate of NEEM Project Members.

As mentioned above, this has been done both in the abstract and main text (see yellow highlights at the end of the document).

As I already suggested, you really only show in this section that we have no good basis for estimating the accumulation rate that applies to this warm interglacial. Here I just comment on some of the estimates you give: i) The Kapsner paper quite conclusively shows that the thermodynamic approach is not really applicable to a situation like that in Greenland, where most of the precipitation is related to cyclones and storm activity. It's fine to mention it as an option, but it is clearly flawed. When it comes to figure 3, I really don't understand what the grey shaded area is meant to represent, so if you leave it, it needs a better explanation. ii) It seems as if this Benson approach is actually an empirical spatial version of the thermodynamic approach? However, while I could almost justify this approach for single storms tracked (and their integrated effect over a year) spatially across Greenland, there seems no basis for translating that into the temporal domain, as you acknowledge on Page 8, line 22. Note also that Benson 1962 seems missing from the reference list.

For simplicity, we have removed the Benson 1962 outputs (indeed a temporal relationship) from Figure 3. The idea is still to keep it as a reference because it is still used today for validation of accumulation rate estimate in Greenland.

 iii) Same applies for the Buchhardt approach, but since the blue lines are missing from the plot I cannot assess it. I understand why you prefer the sensitivity Buchardt derived for the NW region but don't understand why you give a range for Greenland of 6.7-9.6%/K, when Buchardt's table gives a value of only 1.5%/K (with an error bar encompassing zero) for the nearby NE region, and even a negative slope for SW Greenland. I do not think you have summarised the Buchhardt outcome fairly; in my opinion it gives weak evidence for a 6.7%/K slope, but with an uncertainty that encompasses a zero slope. Note also that it is unnecessary to draw a curve to represent Buchardt; he actually derived an acc rate-del18O slope, so for a measured change in del18O, you can pinpoint precisely the range of the change in acc rate and draw them as horizontal lines .

We have included the NW region as well in the discussion of section 3.2.1 and simplified figure 3 as mentioned above (only give the range of possible accumulation rate as for 10Be or chemistry).

Part of section 3.2.1 including the reference to Buchardt 2012:
" More recently, Buchardt et al. (2012) followed a similar approach but using numerous ice cores. They used an array of 52 shallow ice cores spanning the last decades to centuries with accumulation rate estimates from annual layer counting on $\delta^{18}O_{ice}$ profiles. They identified different temperature vs accumulation rate relationships from one region to another. In central and north Greenland corresponding to the location of the NEEM, NGRIP, GRIP and GISP2 deep ice cores, the Buchardt approach suggests a sensitivity of 1.5 to 9.4%.°C$^{-1}$ with an uncertainty encompassing zero. This sensitivity is obtained with a $\delta^{18}O$ vs temperature sensitivity slope of 0.67‰.°C$^{-1}$ so that the Kapsner and Buchardt estimates agree on a 0 to 14%.‰$^{-1}$ accumulation rate vs $\delta^{18}O$ sensitivity. "

iv) I have not
checked the MD approach specifically, but note that this is the same paper that derived a much larger del18O-temperature slope and therefore a smaller temperature change than NEEM Project Members, so it seems inconsistent to use the same model simulations to derive the opposite result.

The accumulation – temperature relationships given in Masson-Delmotte et al. (2015) have indeed a very high slope for the recent warming. This is not necessary inconsistent with the high d18O vs temperature slope but we understand that it may lead to unnecessary complications. We thus propose to gather the accumulation vs temperature and d18O vs temperature relationship from same source (model outputs, reanalyses, …) and only give the final accumulation rate vs d18O relationship (hence combining values from Tables 6 and 7 from Masson-Delmotte et al., 2015). This leads to values around 10 %.‰$^{-1}$, hence a 35% change in accumulation rate for an increase in d18O of 3.5‰ at the NEEM deposition site. This again leads to a temperature reconstruction for the NEEM deposition site at the upper boundary of the 6-11°C range.

Corrected text in section 3.2.1:
"iv: Masson-Delmotte et al. (2015) used estimates of snow accumulation rate and ice $\delta^{18}O$ in four shallow cores in the NEEM area together with accumulation rate, temperature and when possible snowfall $\delta^{18}O$ reconstructions from different models simulation (ECHAM5, Global Climate Model developed by the Max Planck Institute for Meteorology; LMDZ, Global Climate Model developed by the Laboratoire de Météorologie Dynamique; MAR, Regional Atmosphere Model) nudged to available climate reanalyses over the 1979 – 2007 period. In addition to model outputs or temperature gridded reconstruction (Box et al., 2009), the amplitude of temperature increase at NEEM can also be estimated using borehole temperature measurements. Gathering the different sources of information for the strong warming period of 1979-2007 leads to a relatively high slope between accumulation rate and temperature (10 to 15.9 %.°C$^{-1}$, the highest value being obtained using outputs from the MAR model nudged to ERA-40 and ERA-Interim reanalyses (Uppala et al., 2005; Dee et al., 2011)). In this study, the sensitivity of accumulation rate vs. $\delta^{18}O$ can be estimated through the regression between the NEEM $\delta^{18}O$ and accumulation rate increases over the period 1979 – 2007 leading to a value of 10 %.‰$^{-1}$, in agreement with the Buchardt estimate. Another solution is to use the $\delta^{18}O$ vs temperature estimate based on NEEM $\delta^{18}O$ measurements vs borehole temperature over the recent warming trend (0.8‰.°C$^{-1}$) together with the accumulation rate vs temperature

estimate given above, hence leading to a maximum accumulation rate vs $\delta^{18}O$ sensitivity of 13 %.‰$^{-1}$, again within the range of the Buchardt estimate .  "

 v) I have not checked the 10Be estimates, but note
only that because deposition in Greenland under interglacial conditions is completely
dominated by wet deposition, it seems very unlikely that one can derive accumulation
rates from 10Be. I agree with your implication that the change in overall scavenging
would have to be accounted for in a sophisticated model, and I don't believe that local
accumulation rate would be the dominant control on 10Be concentration or flux. For
the same reason, I do not understand at all your statement on Page 9, lines 22-24
that changes in aerosol rule out no change in accumulation: if you want to make such
a statement it needs considerably more explanation and analysis, as there seems no
basis to estimate the changes in sources well enough to make such a statement.

The argument for the statement that the aerosol chemistry renders the same accumulation in the Eemian and the Holocene as unlikely is as follows: If we compare aerosol species which are dominated by wet deposition such as Na (sea salt aerosol) and NO3 (lightning activity, biological activity) we see that the concentration in the ice in the Eemian is drastically lower than in the HOL (only about 50%). As these species are mainly wet deposited this is only possible if the atmospheric aerosol concentrations over the ice sheet was also lower by 50% at that time. It is unlikely that both Na and NO3 (which have completely different sources and transport pathways) have both a 50% reduction of source emissions, which is too large anyway. We can also get a reduction of 50% in atmospheric Na and NO3 concentrations over the ice, if we increase the precipitation rate along the transport pathway and thus increase wet deposition en route. So what the chemistry points evidences is that the precipitation rate during transport in the Northern Hemisphere was significantly higher in the Eemian than in the HOL. If this is not case on the Greenland ice sheet, it means that the precipitation rate should be higher in the Eemian everywhere else except in Greenland, which is unlikely. In particular as sea salt aerosol transport from the open ocean comes jointly with water vapor transport to the ice sheet through storm events an increased washout of Na en route is expected due to higher precipitation rates and this would also dump more water on the ice sheet.
Finally, note that if we scale the change in precipitation rate en route and in local accumulation rate from the HOL to the Eemian according to the Buchardt formula, no change in Na and NO3 sources strength is required at all.

The text has been rewritten:
"v: Alternative estimates of accumulation rate at the NEEM deposition site are also provided by $^{10}$Be data. Sturevik-

Storm et al. (2014) compiled mean Holocene $^{10}$Be concentration over eight Greenland sites and determined the

spatial relationship against mean accumulation rate estimates. This relationship was then applied to LIG $^{10}$Be data

from NEEM, leading to the conclusion that accumulation rate was 65-90% higher than today at the NEEM

deposition site ("$^{10}$Be-approach"). However, the present-day spatial gradients in $^{10}$Be concentration are caused by

spatially varying contributions of wet deposition to the overall $^{10}$Be deposition, assuming a homogeneous

atmospheric $^{10}$Be aerosol deposition over Greenland. This latter assumption implies that the atmosphere above

Greenland is well mixed with respect to [10]Be after transport from the location of troposphere/stratosphere foldings. The latter are the main entrance pathways of stratospheric aerosols into the extratropical northern hemisphere troposphere. The LIG climate is characterized not only by likely enhanced precipitation above Greenland (accumulation) but also higher wet deposition during aerosol transport to Greenland due to higher precipitation rates. As a result, higher scavenging of [10]Be bearing aerosol en route must have led also to a lower atmospheric [10]Be concentration over Greenland than today. The LIG accumulation estimate by Sturevik-Storm et al. (2014) is therefore most likely an overestimation and the assumption of Sturevik-Storm et al. (2014) that [10]Be concentration is only controlled by accumulation rate at the NEEM site may be challenged. More generally, the use of other chemical aerosol species as accumulation rate tracers is hampered by potential changes in the LIG atmospheric concentrations due to emission changes. Qualitatively, a correction of deposition effects using the Buchardt et al. (2012) approach representative for Northwest Greenland, leads to LIG atmospheric concentrations of all chemical aerosol tracers similar to today. In contrast much higher LIG accumulation rates as estimated by Sturevik-Storm et al. (2014) or no changes in accumulation between the LIG and the Holocene imply an unrealistic change in atmospheric aerosol concentrations for several aerosol tracers. Indeed, if we compare aerosol species that are dominated by wet deposition such as $Na^+$ (sea salt aerosol) and $NO_3^-$ (lightning activity, biological activity), we see that the concentration in the ice in the LIG is lower than in the Holocene (about 50%). As these species are mainly wet deposited, this is only possible if the atmospheric aerosol concentration was also reduced by 50% at that time. It is unlikely that both $Na^+$ and $NO_3^-$ (which have completely different sources and transport pathways) have a 50% reduction of source emissions. Another solution to explain the reduction of 50% in atmospheric $Na^+$ and $NO_3^-$ concentration over the ice is to imply an increase of the precipitation rate along the transport pathway and thus increase of wet deposition en route. The chemistry suggests that the precipitation rate during transport in the Northern Hemisphere was significantly higher during the LIG than during the Holocene and there is no reason why Greenland would not be affected by this general increased accumulation rate. Based on the changes in various chemical tracers in the ice (sea salt aerosol, biogenic aerosol, mineral dust) we thus conclude that the LIG accumulation was likely 20 % higher than pre-industrial and similar to the Buchardt approach for Northwest Greenland."

vi) It
seems that GCMs agree with a small change in accumulation rate.
Taking all these estimates together with the one I derived from Kapsner et al, and noting that they all have significant weaknesses, I see no reason to rule out a very small change in accumulation rate, which is suggested by Kapsner et al, approach vi, and the lower range of the Buchardt approach when you include their NE data. It's unfortunate but it means you simply can't constrain accumulation rate this way.

This is correct. It is now included in section 3.2.1 (see above and in the text at the end of the document).

Page 14, line 28. I don't see the relevance of this: at current mean summer temperatures we know we can get melt layers (as we did this year). You certainly don't need the summer mean to be 5 degrees warmer than present to expect significant melt every year.

We were referring to comparison with pre-industrial temperature. In the recent years, melt-layers were observed at NEEM during heat wave as in 2012 with summer temperature of ~5°C warmer than pre-industrial summers.

Fig 1. Please remove the age scale for the shaded portion beyond 128 ka, as we have no basis for it. Summer insolation should also be cut at 128 ka as it cannot be compared to the climate curves below it beyond that age.

Done

Figure 3. Please make this figure clearer in the caption. I suggest something like: "The black circles. . ..NEEM. The light curves are contours of accumulation rate-temperature combinations for a given value if del15N, with that value shown in the curve. The coloured examples correspond to the measured LIG del15N for GRIP (blue), NGRIP (purple) and NEEM (red). The horizontal lines and darker curves correspond to different estimates of the accumulation rate (for the curves, this is as a function of temperature)." Then having explained the overall point of the diagram, you can explain each curve, but of course I would hope that the Buchardt lines will be shown and will include a wider range than in the text, that the Kapsner line is added, and that the vertical arrows better reflect the range of credible estimates.

The figure 3 has been entirely redrawn.

[revised manuscript text omitted]